# Individual Regret in Cooperative Nonstochastic Multi-Armed Bandits

**Yogev Bar-On**
Tel Aviv University, Israel
baronyogev@gmail.com

**Yishay Mansour**
Tel Aviv University, Israel
and Google Research, Israel
mansour.yishay@gmail.com

## Abstract

We study agents communicating over an underlying network by exchanging messages, in order to optimize their individual regret in a common nonstochastic multi-armed bandit problem. We derive regret minimization algorithms that guarantee for each agent $v$ an individual expected regret of $\widetilde{O}\left(\sqrt{\left(1 + \frac{K}{|\mathcal{N}(v)|}\right)T}\right)$, where $T$ is the number of time steps, $K$ is the number of actions and $\mathcal{N}(v)$ is the set of neighbors of agent $v$ in the communication graph. We present algorithms both for the case that the communication graph is known to all the agents, and for the case that the graph is unknown. When the graph is unknown, each agent knows only the set of its neighbors and an upper bound on the total number of agents. The individual regret between the models differs only by a logarithmic factor. Our work resolves an open problem from [Cesa-Bianchi et al., 2019b].

## 1 Introduction

The multi-armed bandit (MAB) problem is one of the most basic models for decision making under uncertainty. It highlights the agent's uncertainty regarding the losses it suffers from selecting various actions. The agent selects actions in an online fashion - each time step the agent selects a single action and suffers a loss corresponding to that action. The agent's goal is to minimize its cumulative loss over a fixed horizon of time steps. The agent observes only the loss of the action it selected each step. Therefore, the MAB problem captures well the crucial trade-off between exploration and exploitation, where the agent needs to explore various actions in order to gather information about them.

MAB research discusses two main settings: the stochastic setting, where the losses of each action are sampled i.i.d. from an unknown distribution, and the nonstochastic (adversarial) setting, where we make no assumptions about the loss sequences. In this work we consider the nonstochastic setting and the objective of minimizing the regret - the difference between the agent's cumulative loss and the cumulative loss of the best action in hindsight. It is known that a regret of the order of $\Theta\left(\sqrt{KT}\right)$ is the best that can be guaranteed, where $K$ is the number of actions and $T$ is the time horizon. In contrast, when the losses of all actions are observed (full-information feedback) the regret can be of the order of $\Theta\left(\sqrt{T \ln K}\right)$ (see, e.g., [Cesa-Bianchi and Lugosi, 2006, Bubeck et al., 2012]).

The main focus of our work is to consider agents that are connected in a communication graph, and can exchange messages in each step, in order to reduce their individual regret. This is possible since the losses depend only on the action and the time step, but not on the agent.

One extreme case is when the communication graph is a clique, i.e., any pair of agents can communicate directly. In this case, the agents can run the well known Exp3 algorithm [Auer et al., 2002], and guarantee each a regret of $O\left(\sqrt{T \ln K}\right)$, assuming there are at least $K$ agents (see [Seldin et al.,

2014, Cesa-Bianchi et al., 2019b]). However, in many motivating applications, such as distributed learning, or communication tasks such as routing, the communication graph is not a clique.

The work of Cesa-Bianchi et al. [2019b] studies a general communication graph, where the agents can communicate in order to reduce their regret. The paper presents the Exp3-Coop algorithm, which achieves an expected regret **when averaged over all agents** of $\widetilde{O}\left(\sqrt{\left(1 + \frac{K}{N}\alpha\left(G\right)\right)T}\right)$, where $\alpha\left(G\right)$ is the independence number of the communication graph $G$, and $N$ is the number of agents. The question of whether it is possible to obtain a low **individual regret**, that holds simultaneously for all agents, was left as an open question. We answer this question affirmatively in this work.

Our main contribution is an individual expected regret bound, which holds for each agent $v$, of order

$$\widetilde{O}\left(\sqrt{\left(1 + \frac{K}{|\mathcal{N}\left(v\right)|}\right)T}\right),$$

where $\mathcal{N}\left(v\right)$ is the set of neighbors of agent $v$ in the communication graph. We remark that our result also implies the previous average regret bound.

The main idea of our algorithm is to artificially partition the graph into disjoint connected components. Each component has a center agent, which is in some sense the leader of the component. The center agent has (almost) the largest degree in the component, and it selects actions using the Exp3-Coop algorithm. By observing the outcomes of its immediate neighboring agents, the center agent can guarantee its own desired individual regret. The main challenge is to create such components with a relatively small diameter, so that the center will be able to broadcast its information in a short time to all the agents in the component. Special care is given to relate the agents' local parameters (degree) to the global component parameters (degree of the center agent and the broadcast time).

We consider both the case that the communication graph is known to all the agents in advance (the informed setting), and the case that the graph is unknown (the uninformed setting). In the uninformed setting, we assume each agent knows its local neighborhood (i.e., the set of its neighbors), and an upper bound on the total number of agents. The regret bound in the uninformed setting is higher by a logarithmic factor and the algorithm is more complex.

In the next section, we formally define our model, and review preliminary material. Section 3 shows the center-based policy, given a graph partition. We then present our graph partitioning algorithms in Section 4. Overview of the analysis is given in Section 5, while all proofs are differed to the supplementary material. Our work is concluded in Section 6.

## 1.1 Additional related works

The cooperative nonstochastic MAB setting was introduced by Awerbuch and Kleinberg [2008], where they bound the average regret, when some agents might be dishonest and the communication is done through a public channel (clique network). The previously mentioned [Cesa-Bianchi et al., 2019b], also considers the issue of delays, and presents a bound on the average regret for a general graph of order $\widetilde{O}\left(\sqrt{\left(d + \frac{K}{N}\alpha\left(G\right)\right)T} + d\right)$, when messages need $d$ steps to arrive. Dist-Hedge, introduced by Sahu and Kar [2017], considers a network of forecasting agents, with delayed and inexact losses, and derives a sub-linear individual regret bound, that also depends on spectral properties of the graph. More recently, Cesa-Bianchi et al. [2019a] studied an online learning model where only a subset of the agents play at each time step, and showed matching upper and lower bounds on the average regret of order $\sqrt{\alpha\left(G\right)T}$ when the set of agents that play each step is chosen stochastically. When the set of agents is chosen arbitrarily, the lower bound becomes $T$.

In the stochastic setting, Landgren et al. [2016a,b] presented a cooperative variant of the well-known UCB algorithm, that uses a consensus algorithm for estimating the mean losses, to obtain a low average regret. More cooperative variants of the UCB algorithm that yield a low average regret were presented by Kolla et al. [2018]. They also showed a policy, where like in the methods in this work, agents with a low degree follow the actions of agents with a high degree. Stochastic MAB over P2P communication networks were studied by Szörényi et al. [2013], which showed that the probability to select a sub-optimal arm reduces linearly with the number of peers. The case where only one agent can observe losses was investigated by Kar et al. [2011]. This agent needs to broadcast information through the network, and it was shown this is enough to obtain a low average regret.

Another multi-agent research area involve agents that compete on shared resources. The motivation comes from radio channel selection, where multiple devices need to choose a radio channel, and two or more devices that use the same channel simultaneously interfere with each other. In this setting, many papers assume agents cannot communicate with each other, and do not receive a reward upon collision - where more than one agent tries to choose the same action at the same step. The first to give regret bounds on this variant are Avner and Mannor [2014], that presented an average regret bound of order $O\left(T^{\frac{2}{3}}\right)$ in the stochastic setting. Also in the stochastic setting, Rosenski et al. [2016] showed an expected average regret bound of order $O\left(\frac{K}{\Delta^2}\ln\left(\frac{K}{\delta}\right)+N\right)$ that holds with probability $1-\delta$, where $\Delta$ is the minimal gap between the mean rewards (notice that this bound is independent of $T$). In the same paper, they also studied the case that the number of agents may change each step, and presented a regret bound of $\widetilde{O}\left(\sqrt{xT}\right)$, where $x$ is the total number of agents throughout the game. Bistritz and Leshem [2018] consider the case that different agents have different mean rewards, and each agent has a different unique action it should choose to maximize the total regret. They showed an average regret of order $O\left(\log^{2+\epsilon}T\right)$ for every $\epsilon>0$, where the $O$-notation hides the dependency on the mean rewards.

## 2   Preliminaries

We consider a nonstochastic multi-armed bandit problem over a finite action set $A=\{1,\ldots,K\}$ played by $N$ agents. Let $G=\langle V,E\rangle$ be an undirected connected communication graph for the set of agents $V=\{1,\ldots,N\}$, and denote by $\mathcal{N}(v)$ the neighborhood of $v\in V$, including itself. Namely,

$$\mathcal{N}(v)=\{u\in V\mid\langle u,v\rangle\in E\}\cup\{v\}.$$

At each time step $t=1,2,\ldots,T$, each agent $v\in V$ draws an action $I_t(v)\in A$ from a distribution $\boldsymbol{p}_t^v=\langle p_t^v(1),\ldots,p_t^v(K)\rangle$ on $A$. It then suffers a loss $\ell_t(I_t(v))\in[0,1]$ which it observes. Notice the loss does not depend on the agent, but only on the time step and the chosen action. Thus, agents that pick the same action at the same step will suffer the same loss. We also assume the adversary is oblivious, i.e., the losses do not depend on the agents' realized actions. In the end of step $t$, each agent sends a message

$$m_t(v)=\langle v,t,I_t(v),\ell_t(I_t(v)),\boldsymbol{p}_t^v\rangle$$

to all the agents in its neighborhood, and also receives messages from its neighbors: $m_t(v')$ for all $v'\in\mathcal{N}(v)$. Our goal is to minimize, for each $v\in V$, its *expected regret* over $T$ steps:

$$R_T(v)=\mathbb{E}\left[\sum_{t=1}^T\ell_t(I_t(v))-\min_{i\in A}\sum_{t=1}^T\ell_t(i)\right].$$

A well-known policy to update $\boldsymbol{p}_t^v$ is the exponential-weights algorithm (Exp3) with weights $w_t^v(i)$ for all $i\in A$, such that $p_t^v(i)=\frac{w_t^v(i)}{W_t^v}$ where $W_t^v=\sum_{i\in A}w_t^v(i)$ (see, e.g., [Cesa-Bianchi and Lugosi, 2006]). The weights are updated as follows: let $B_t^v(i)$ be the event that $v$ observed the loss of action $i$ at step $t$; in our case $B_t^v(i)=\mathbb{I}\{\exists v'\in\mathcal{N}(v):I_t(v')=i\}$, where $\mathbb{I}$ is the indicator function. Also, let $\hat{\ell}_t^v(i)=\frac{\ell_t(i)}{\mathbb{E}_t[B_t^v(i)]}B_t^v(i)$ be an unbiased estimated loss of action $i$ at step $t$, where $\mathbb{E}_t[\cdot]$ is the expectation conditioned on all the agents' choices up to step $t$ (hence, $\mathbb{E}_t\left[\hat{\ell}_t^v(i)\right]=\ell_t(i)$). Then

$$w_{t+1}^v(i)=w_t^v(i)\exp\left(-\eta(v)\hat{\ell}_t^v(i)\right),$$

where $\eta(v)$ is a positive parameter chosen by $v$, called the *learning rate* of agent $v$. Exp3 is given explicitly in the supplementary material. Notice that in our setting all agents $v\in V$ have the information needed to compute $\hat{\ell}_t^v(i)$, since

$$\mathbb{E}_t[B_t^v(i)]=\Pr[\exists v'\in\mathcal{N}(v):I_t(v')=i]=1-\prod_{v'\in\mathcal{N}(v)}\left(1-p_t^{v'}(i)\right),$$

and if agent $v$ does not observe $\ell_t(i)$, then $\hat{\ell}_t^v(i)=0$.

We proceed with two useful lemmas that will help us later. For completeness, we provide their proofs in the supplementary material as well. The first lemma is the usual analysis of the exponential-weights algorithm:

**Lemma 1.** *Assuming agent $v$ uses the exponential-weights algorithm, its expected regret satisfies*

$$R_T(v) \leq \frac{\ln K}{\eta(v)} + \frac{\eta(v)}{2} \mathbb{E}\left[\sum_{t=1}^{T}\sum_{i=1}^{K} p_t^v(i)\, \hat{\ell}_t^v(i)^2\right].$$

The next lemma is from [Cesa-Bianchi et al., 2019b], and it bounds the change of the action distribution in the exponential-weights algorithm.

**Lemma 2.** *Assuming agent $v$ uses the exponential-weights algorithm with a learning rate $\eta(v) \leq \frac{1}{2K}$, then for all $i \in A$:*

$$\left(1 - \eta(v)\,\hat{\ell}_t^v(i)\right) p_t^v(i) \leq p_{t+1}^v(i) \leq 2p_t^v(i).$$

Also, the following definition will be needed for our algorithm. We denote by $G^r$ the $r$-th power of $G$, in which $v_1, v_2 \in V$ are adjacent if and only if $\mathrm{dist}_G(v, v') \leq r$; and by $G_{|U}$ the sub-graph of $G$ induced by $U \subseteq V$.

**Definition 3.** Let $G = \langle V, E \rangle$ be an undirected connected graph and let $W \subseteq U \subseteq V$. $W$ is called an *$r$-independent set* of $G$, if it is an independent set of $G^r$. Namely,

$$\forall w, w' \in W : \mathrm{dist}_G(w, w') \geq r + 1.$$

If $W$ is also a maximal independent set of $(G^r)_{|U}$, it is called a *maximal $r$-independent subset* ($r$-MIS) of $U$. Namely, there is no $r$-independent set $W' \subseteq U$ such that $W \subset W'$.

## 3 Center-based cooperative multi-armed bandits

We now present the center-based policy for the cooperative multi-armed bandit setting, which will give us the desired low individual regret. In the center-based cooperative MAB, not all the agents behave similarly. We partition the agents to three different types.

*Center agents* are the agents that determine the action distribution for all other agents. They work together with their neighbors to minimize their regret. The neighbors of the center agents in the communication graph, *center-adjacent* agents, always copy the action distribution from their neighboring center, and thus the centers gain more information about their own distribution each step.

Other (not center or center-adjacent) agents are *simple agents*, which simply copy the action distribution from one of the centers. Since they are not center-adjacent, they receive the action distribution with delay, through other agents that copy from the same center.

We artificially partition the graph to connected components, such that each center $c$ has its own component, and all the simple agents in the component of $c$ copy their action distribution from it. To obtain a low individual regret, we require the components to have a relatively small diameter, and the center agents to have a high degree in the communication graph. Namely, center agents have the highest or nearly highest degree in their component.

In more detail, we select a set $C \subseteq V$ of center agents. All center agents $c \in C$ use the exponential-weights algorithm with a learning rate $\eta(c) = \frac{1}{2}\sqrt{\frac{(\ln K)\min\{|\mathcal{N}(c)|, K\}}{KT}}$. The agent set $V$ is partitioned into disjoint subsets $\{V_c \subseteq V \mid c \in C\}$, such that $\mathcal{N}(c) \subseteq V_c$ for all $c \in C$, and the sub-graph $G_c \equiv G_{|V_c}$ induced by $V_c$ is connected. Notice that since the components are disjoint, the condition $\mathcal{N}(c) \subseteq V_c$ implies $C$ is a 2-independent set. For all non-centers $v \in V \setminus C$, we denote by $\mathcal{C}(v) \in C$ the center agent such that $v \in V_{\mathcal{C}(v)}$, and call it the *center of $v$*. All non-center agents $v \in V \setminus C$ copy their distribution from their *origin neighbor $U(v)$*, which is their neighbor in $G_{\mathcal{C}(v)}$ closest to $\mathcal{C}(v)$, breaking ties arbitrarily. Namely,

$$U(v) = \underset{v' \in \mathcal{N}(v) \cap V_{\mathcal{C}(v)}}{\arg\min} \mathrm{dist}_{G_{\mathcal{C}(v)}}(v', \mathcal{C}(v)).$$

Thus, agent $v$ receives its center's distribution with a delay of $d(v) = \mathrm{dist}_{G_{\mathcal{C}(v)}}(v, \mathcal{C}(v))$ steps, so for all $t \geq d(v) + 1$:

$$\boldsymbol{p}_t^v = \boldsymbol{p}_{t-d(v)}^{\mathcal{C}(v)}.$$

Notice that if $v \in \mathcal{N}(c)$, then $v$ is center-adjacent and it holds $U(v) = \mathcal{C}(v)$ and $d(v) = 1$. For completeness, we define $U(c) = \mathcal{C}(c) = c$ and $d(c) = 0$ for all $c \in C$.

To express the regret of the center-based policy, we introduce a new concept:

---
**Algorithm 1** Center-based cooperative MAB - $v$ is a center agent
---
**Parameters:** Number of arms $K$; Time horizon $T$.

**Initialize:** $\eta\left(v\right) \leftarrow \frac{1}{2}\sqrt{\frac{(\ln K)M(v)}{KT}}$; $w_1^v\left(i\right) \leftarrow \frac{1}{K}$ for all $i \in A$.

1: **for** $t \leq T$ **do**
2:      Set $p_t^v\left(i\right) \leftarrow \frac{w_t^v(i)}{W_t^v}$ for all $i \in A$, where $W_t^v = \sum_{i \in A} w_t^v\left(i\right)$.
3:      Play an action $I_t\left(v\right)$ drawn from $\boldsymbol{p}_t^v = \langle p_t^v\left(1\right), \ldots, p_t^v\left(K\right)\rangle$.
4:      Observe loss $\ell_t\left(I_t\left(v\right)\right)$.
5:      Send the following message to the set $\mathcal{N}\left(v\right)$: $m_t\left(v\right) = \langle v, t, I_t\left(v\right), \ell_t\left(I_t\left(v\right)\right), \boldsymbol{p}_t^v\rangle$.
6:      Receive all messages $m_t\left(v'\right)$ from $v' \in \mathcal{N}\left(v\right)$.
7:      Update for all $i \in A$: $w_{t+1}^v\left(i\right) \leftarrow w_t^v\left(i\right)\exp\left(-\eta\left(v\right)\hat{\ell}_t^v\left(i\right)\right)$, where

$$\hat{\ell}_t^v\left(i\right) = \frac{\ell_t\left(i\right)}{\mathbb{E}_t\left[B_t^v\left(i\right)\right]}B_t^v\left(i\right),$$

$$B_t^v\left(i\right) = \mathbb{I}\left\{\exists v' \in \mathcal{N}\left(v\right) : I_t\left(v'\right) = i\right\}, \quad \mathbb{E}_t\left[B_t^v\left(i\right)\right] = 1 - \prod_{v' \in \mathcal{N}(v)}\left(1 - p_t^{v'}\left(i\right)\right).$$

8: **end for**
---

---
**Algorithm 2** Center-based cooperative MAB - $v$ is a non-center agent
---
**Parameters:** Number of arms $K$; Time horizon $T$; Origin neighbor $U\left(v\right)$.

**Initialize:** $p_1^v\left(i\right) \leftarrow \frac{1}{K}$ for all $i \in A$.

1: **for** $t \leq T$ **do**
2:      Play an action $I_t\left(v\right)$ drawn from $\boldsymbol{p}_t^v = \langle p_t^v\left(1\right), \ldots, p_t^v\left(K\right)\rangle$.
3:      Observe loss $\ell_t\left(I_t\left(v\right)\right)$.
4:      Send the following message to the set $\mathcal{N}\left(v\right)$: $m_t\left(v\right) = \langle v, t, I_t\left(v\right), \ell_t\left(I_t\left(v\right)\right), \boldsymbol{p}_t^v\rangle$.
5:      Receive the message $m_t\left(U\left(v\right)\right)$ from $U\left(v\right)$.
6:      Update $p_{t+1}^v\left(i\right) = p_t^{U(v)}\left(i\right)$ for all $i \in A$.
7: **end for**
---

**Definition 4.** The *mass* of a center agent $c \in C$ is defined to be
$$M\left(c\right) \equiv \min\left\{\left|\mathcal{N}\left(c\right)\right|, K\right\},$$
and the mass of non-center agent $v \in V \setminus C$ is
$$M\left(v\right) \equiv e^{-\frac{1}{6}d(v)}M\left(\mathcal{C}\left(v\right)\right).$$

Notice the mass depends only on how the graph is partitioned, and it satisfies $M\left(v\right) = e^{-\frac{1}{6}}M\left(U\left(v\right)\right)$ for all non-centers $v \in V \setminus C$. Intuitively, the mass of agent $v$ captures the idea that as the degree of the center is larger and as the agent is closer to its center, the lower the regret of $v$. We prove that the regret is $\widetilde{O}\left(\sqrt{\frac{K}{M(v)}T}\right)$. Our partitioning algorithms, presented in the next section, show that the mass of agent $v$ satisfies $M\left(v\right) = \Omega\left(\min\left\{\left|\mathcal{N}\left(v\right)\right|, K\right\}\right)$, so we obtain an individual regret of the order of $\widetilde{O}\left(\sqrt{\left(1 + \frac{K}{|\mathcal{N}(v)|}\right)T}\right)$.

We specify the center-based policy in Algorithms 1 and 2. We emphasize that before the agents use the center-based policy they must partition the graph with one of the algorithms we present in the next section. While the agents partition the graph, they play arbitrary actions.

## 4 Partitioning the graph

The goal now is to show that we can partition the graph such that the mass is large for every $v \in V$. In particular, we want to show that any graph can be partitioned such that $M\left(v\right) = \Omega\left(\min\left\{\left|\mathcal{N}\left(v\right)\right|, K\right\}\right)$.

We consider two cases: the informed and uninformed settings. In the informed setting, all of the agents have access to the graph structure. Each agent can partition the graph by itself in advance,

---

**Algorithm 3** Centers-to-Components

---

**Parameters:** Number of arms $K$; Center set $C$.

**Initialize:** Number of iterations $\Theta_K \leftarrow \lfloor 12 \ln K \rfloor$.

1: **if** $v \in C$ **then**
2:     Initialize: $\mathcal{C}_0(v) \leftarrow v; \quad U_0(v) \leftarrow v; \quad M_0(v) \leftarrow \min\{|\mathcal{N}(v)|, K\}$.
3: **else**
4:     Initialize: $\mathcal{C}_0(v) \leftarrow \text{nil}; \quad U_0(v) \leftarrow \text{nil}; \quad M_0(v) \leftarrow 0$.
5: **end if**
6: **for** $0 \leq t \leq \Theta_K$ **do**
7:     Send the following message to the set $\mathcal{N}(v)$: $\mu_t(v) = \langle v, t, \mathcal{C}_t(v), M_t(v) \rangle$.
8:     Receive all messages $\mu_t(v')$ from $v' \in \mathcal{N}(v)$.
9:     **if** $U_t(v) \notin C$ **then**         $\triangleright$ The center-based policy requires $\mathcal{N}(c) \subseteq V_c$ for all $c \in C$.
10:         Find the best origin neighbor for $v$:

$$U_{t+1}(v) \leftarrow \underset{v' \in \mathcal{N}(v) \setminus \{v\}}{\arg\max} \ M_t(v').$$

11:         Update: $\mathcal{C}_{t+1}(v) \leftarrow \mathcal{C}_t(U_{t+1}(v)); \quad M_{t+1}(v) \leftarrow e^{-\frac{1}{6}} M_t(U_{t+1}(v))$.
12:     **else**
13:         Keep old values: $\mathcal{C}_{t+1}(v) \leftarrow \mathcal{C}_t(v); \quad U_{t+1}(v) \leftarrow U_t(v); \quad M_{t+1}(v) \leftarrow M_t(v)$.
14:     **end if**
15: **end for**
16: **return**

$$\mathcal{C}(v) = \mathcal{C}_{\Theta_K + 1}(v); \quad U(v) = U_{\Theta_K + 1}(v); \quad M(v) = M_{\Theta_K + 1}(v).$$

---

to know the role it plays: whether it is a center or not, and which agent is its origin neighbor. In the uninformed setting, the graph structure is not known to the agents, only their neighbors and an upper bound on the total number of agents $\bar{N} \geq N$. The agents partition the graph using a distributed algorithm while playing actions and suffering loss.

The basic structure of the partitioning algorithm in both settings is the same. First, we show an algorithm that computes the connected components given a center set $C$. Then, we show an algorithm that computes a center set $C$. The second algorithm is specifically designed to be used with the first, and together they partition the graph to connected components such that every agent has a large mass.

## 4.1 Computing graph components given a center set

Given a center set $C$, we show a distributed algorithm called *Centers-to-Components*, which computes the connected components, and present it in Algorithm 3. Although it is distributed, in the informed setting agents can simply simulate it locally in advance.

Centers-to-Components runs simultaneous distributed BFS graph traversals, originating from every center $c \in C$. When the traversal of center $c$ arrives to a simple agent $v \in V \setminus C$, $v$ decides if $c$ is the best center for it so far, and if it is, $v$ switches its component to $V_c$. Notice each agent needs to know only if itself is a center or not.

## 4.2 Computing centers

To compute the center set $C$, we show two algorithms; one for the informed setting and one for the uninformed setting. The regret bound for the informed setting is slightly better, and the algorithm is simpler.

**The informed setting** The algorithm that computes the center set in the informed setting is called *Compute-Centers-Informed* and is presented in Algorithm 4. The center set is built in a greedy way: each iteration, all of the agents test if they are "satisfied" with the current center set (i.e., $M(v) \geq \min\{|\mathcal{N}(v)|, K\}$). If there are unsatisfied agents left, the agent with the highest degree is added to the center set.

---

**Algorithm 4** Compute-Centers-Informed

---

**Parameters:** Undirected connected graph $G = \langle V, E \rangle$; Number of arms $K$.
**Initialize:** Center set $C_0 \leftarrow \emptyset$; Unsatisfied agents $S_0 \leftarrow V$.

 1: $t \leftarrow 0$.
 2: **while** $S_t \neq \emptyset$ **do**
 3:      Choose the next center: $c_t \leftarrow \arg\max_{v \in S_t} |\mathcal{N}(v)|$.
 4:      Update $C_{t+1} \leftarrow C_t \cup \{c_t\}$.
 5:      Run Centers-to-Components with center set $C_{t+1}$, and obtain mass $M_{t+1}(v)$ for each $v \in V$.
 6:      Update

$$S_{t+1} \leftarrow \left\{ v \in V \mid M_{t+1}(v) < \min\{|\mathcal{N}(v)|, K\} \wedge \min_{c \in C_{t+1}} \operatorname{dist}_G(v, c) \geq 3 \right\}.$$

 7:      $t \leftarrow t + 1$.
 8: **end while**
 9: **return** $C = C_t$.

---

**The uninformed setting** At first, it may seem that the uninformed setting can be solved the same way as the informed setting, with some distributed version of Compute-Centers-Informed. However, such algorithm will require $\Omega(N)$ steps in the worst case, since at each iteration only one agent becomes a center. In the informed setting we do not care about this, since the components are computed in advance. In the uninformed setting however, at each step of the algorithm the agents suffer a loss, and thus the regret bound will be at least linear in the number of agents, which can be very large.

To avoid this problem, we need to add many centers each iteration, and not just one as in Compute-Centers-Informed. To do this, we exploit the fact that there are only $K$ possible values for a center's mass. In our algorithm, there are $K$ iterations, and in each iteration $t$, as many agents as possible with degree $K - t$ become centers. To ensure the final center set is 2-independent, only a 2-MIS of the potential center agents are added to the center set each iteration.

To compute a 2-MIS in a distributed manner, we use Luby's algorithm [Luby, 1986, Alon et al., 1986] on the sub-graph of $G^2$ induced by the potential center agents. Briefly, at each iteration of Luby's algorithm, every potential center agent picks a number uniformly from $[0, 1]$. Agents that picked the maximal number among their neighbors of distance 2 join the 2-MIS, and their neighbors of distance 2 stop participating. A 2-MIS is computed after $\left\lceil 3 \ln\left(\frac{N}{\sqrt{\delta}}\right) \right\rceil$ iterations with probability $1 - \delta$. Each iteration requires exchanging 4 messages - 2 for communicating the random numbers and 2 for communicating the new agents in the 2-MIS. Hence, $4 \left\lceil 3 \ln\left(\frac{N}{\sqrt{\delta}}\right) \right\rceil$ steps suffice to compute a 2-MIS with probability $1 - \delta$. A more detailed explanation of Luby's algorithm can be found in the supplementary material.

We present *Compute-Centers-Uninformed* in Algorithm 5. Since this is a distributed algorithm, we have the variables $\mathbb{C}(v)$ and $\mathbb{S}(v)$ as indicators for whether $v$ is a center or unsatisfied, respectively.

## 5 Regret analysis

We will now provide an overview for the analysis of our algorithms. We remind that all proofs are differed to the supplementary material.

### 5.1 Individual regret of the center-based policy

We start by bounding the expected regret of the agents when they are using the center-based policy.

**Theorem 5.** *Let $T \geq K^2 \ln K$. Using the center-based policy, the regret of each agent $v \in V$ satisfies*

$$R_T(v) \leq 7 \sqrt{(\ln K) \frac{K}{M(v)} T}.$$

---

**Algorithm 5** Compute-Centers-Uninformed - agent $v$

---

**Parameters:** Number of arms $K$; Upper bound on the total number of agents $\bar{N}$; Time horizon $T$.
**Initialize:** Center indicator $\mathbb{C}(v) \leftarrow$ FALSE; Unsatisfied indicator $\mathbb{S}(v) \leftarrow$ TRUE.
1: **for** $0 \leq t \leq K - 1$ **do**
2:      Participate for $4 \left\lceil 3 \ln \left( \bar{N} \sqrt{KT} \right) \right\rceil$ steps in Luby's algorithm on $\left( G^2 \right)_{|S_t}$, where

$$S_t = \{ v \in V \mid \mathbb{S}(v) = \text{TRUE} \wedge \min \{ |\mathcal{N}(v)|, K \} = K - t \},$$

   to compute $W_t$, a 2-MIS of $S_t$, with probability $1 - \frac{1}{TK}$.
3:      If $v \in W_t$, set $\mathbb{C}(v) \leftarrow$ TRUE.
4:      Participate in Centers-to-Components with center set $C_t = \{ v' \in V \mid \mathbb{C}(v') = \text{TRUE} \}$;
   obtain mass $M_t(v)$ and whether $\min_{c \in C_t} \text{dist}_G(v, c) \geq 3$.
5:            $\triangleright \min_{c \in C_t} \text{dist}_G(v, c) \geq 3$ if and only if $\mathcal{C}_2(v) = \text{nil}$ in Centers-to-Components.
6:      Update

$$\mathbb{S}(v) \leftarrow \mathbb{I} \left[ M_t(v) < \min \{ |\mathcal{N}(v)|, K \} \wedge \min_{c \in C_t} \text{dist}_G(v, c) \geq 3 \right].$$

7: **end for**
8: **return** $C = C_{K-1}$.

---

This individual regret bound holds simultaneously for all agents in the graph, and it depends only on the graph structure and components.

## 5.2 Analyzing Centers-to-Components

We need to show the results of Centers-to-Components follow their definitions, and the derived components satisfy all the properties required by the center-based policy. The following lemma show it under some requirements from the center set $C$.

**Lemma 6.** *Let $C \subseteq V$ be a center set that is 2-independent, such that every $v \in V$ holds $\min_{c \in C} \text{dist}_G(v, c) \leq 6 \ln K - 1$. Let $\mathcal{C}(v), U(v), M(v)$ be the results of Centers-to-Components. For each $c \in C$, let $V_c$ be its corresponding component, namely, $V_c = \{ v \in V \mid \mathcal{C}(v) = c \}$. Then the following properties are satisfied:*

1. *$\{ V_c \mid c \in C \}$ are pairwise disjoint and $V = \bigcup_{c \in C} V_c$.*

2. *$\mathcal{N}(c) \subseteq V_c$ and $G_c$ is connected for all $c \in C$.*

3. *$M(v) = e^{-\frac{1}{6} d(v)} M(\mathcal{C}(v))$ and $U(v) = \arg\min_{v' \in \mathcal{N}(v) \cap V_{\mathcal{C}(v)}} d(v')$ for all $v \in V \setminus C$.*

## 5.3 Analyzing Compute-Centers-Informed

The first thing we need to show is that the center set returned by Compute-Centers-Informed satisfies the conditions of Lemma 6:

**Lemma 7.** *Let $C \subseteq V$ be the center set returned by Compute-Centers-Informed. Then:*

1. *$C$ is 2-independent.*

2. *For all $v \in V$, $\min_{c \in C} \text{dist}_G(v, c) \leq 6 \ln K - 1$.*

Now, we can show that by using our informed graph partitioning algorithms, the mass of all agents is large:

**Theorem 8.** *Let $C \subseteq V$ be the center set returned by Compute-Centers-Informed, and let $\{ V_c \subseteq V \mid c \in C \}$ be the components resulted from Centers-to-Components. For every $v \in V$:*

$$M(v) \geq e^{-1} \min \{ |\mathcal{N}(v)|, K \}.$$

Together with Theorem 5, we obtain the desired regret bound.

**Corollary 9.** *Let $T \geq K^2 \ln K$. Let $C \subseteq V$ be the center set returned by Compute-Centers-Informed, and let $\{V_c \subseteq V \mid c \in C\}$ be the components resulted from Centers-to-Components. Using the center-based policy, we obtain for every $v \in V$:*

$$R_T(v) \leq 12\sqrt{(\ln K)\left(1 + \frac{K}{|\mathcal{N}(v)|}\right)T} = \widetilde{O}\left(\sqrt{\left(1 + \frac{K}{|\mathcal{N}(v)|}\right)T}\right).$$

### 5.4 Analyzing Compute-Centers-Uninformed

First, we show that Compute-Centers-Uninformed terminates after a relatively small number of steps, and thus the loss suffered while running it is insignificant.

**Lemma 10.** *Compute-Centers-Uninformed runs for less than $12K \ln\left(K^2 \bar{N} T\right)$ steps.*

As in the informed setting, we now need to show the center set resulted from Compute-Centers-Uninformed satisfies the conditions of Lemma 6.

**Lemma 11.** *Let $C \subseteq V$ be the center set resulted from Compute-Centers-Uninformed, such that Luby's algorithm succeeded at all iterations of the algorithm. Then:*

1. *$C$ is 2-independent.*

2. *For all $v \in V$, $\min_{c \in C} \operatorname{dist}_G(v, c) \leq 6 \ln K - 1$.*

We can now obtain the same result as in the informed setting:

**Theorem 12.** *Let $C \subseteq V$ be the center set resulted from Compute-Centers-Uninformed, such that Luby's algorithm succeeded at all iterations of the algorithm, and also let $\{V_c \subseteq V \mid c \in C\}$ be the components resulted from Centers-to-Components. For every $v \in V$:*

$$M(v) \geq e^{-1} \min\{|\mathcal{N}(v)|, K\}.$$

Again we can use Theorem 5 to obtain the desired regret bound.

**Corollary 13.** *Let $T \geq K^2 \ln K$ and $\bar{N} \geq N$. Let $C \subseteq V$ be the center set resulted from Compute-Centers-Uninformed, and let $\{V_c \subseteq V \mid c \in C\}$ be the components resulted from Centers-to-Components. Using the center-based policy, we obtain for every $v \in V$:*

$$R_T(v) \leq 12\left(K \ln\left(K^2 \bar{N} T\right) + \sqrt{(\ln K)\left(1 + \frac{K}{|\mathcal{N}(v)|}\right)T}\right) + 1 = \widetilde{O}\left(\sqrt{\left(1 + \frac{K}{|\mathcal{N}(v)|}\right)T}\right).$$

### 5.5 Average regret of the center-based policy

As mentioned before, we strictly improve the result of Cesa-Bianchi et al. [2019b], and our algorithms imply the same average expected regret bound.

**Corollary 14.** *Let $T \geq K^2 \ln K$. Let $C \subseteq V$ be the center set resulted from Compute-Centers-Informed or Compute-Centers-Uninformed, and let $\{V_c \subseteq V \mid c \in C\}$ be the components resulted from Centers-to-Components. Using the center-based policy, we get:*

$$\frac{1}{N}\sum_{v \in V} R_T(v) = \widetilde{O}\left(\sqrt{\left(1 + \frac{K}{N}\alpha(G)\right)T}\right).$$

## 6 Conclusions

We investigated the cooperative nonstochastic multi-armed bandit problem, and presented the center-based cooperation policy (Algorithms 1 and 2). We provided partitioning algorithms that provably yield a low individual regret bound that holds simultaneously for all agents (Algorithms 3, 4 and 5). We express this bound in terms of the agents' degree in the communication graph. This bound strictly improves a previous regret bound from [Cesa-Bianchi et al., 2019b] (Corollary 14), and also resolves an open question from that paper.

Note that our regret bound in the informed setting does not depend on the total number of agents, $N$, and in the uninformed setting it depends on $\bar{N}$ only logarithmically. It is unclear whether in the uninformed setting, any dependence on $N$ in the individual regret is required.

## Acknowledgments

This work was supported in part by the Yandex Initiative in Machine Learning and by a grant from the Israel Science Foundation (ISF).

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
