\left(v\right) \leq \frac{\ln K}{\eta\left(v\right)} + \frac{\eta\left(v\right)}{2}\mathbb{E}\left[\sum_{t=1}^{T}\sum_{i=1}^{K} p_t^v\left(i\right)\hat{\ell}_t^v\left(i\right)^2\right].$$

The next lemma is from [Cesa-Bianchi et al., 2019b], and it bounds the change of the action distribution in the exponential-weights algorithm.

**Lemma 2.** *Assuming agent $v$ uses the exponential-weights algorithm with a learning rate $\eta\left(v\right) \leq \frac{1}{2K}$, then for all $i \in A$:*

$$\left(1 - \eta\left(v\right)\hat{\ell}_t^v\left(i\right)\right)p_t^v\left(i\right) \leq p_{t+1}^v\left(i\right) \leq 2p_t^v\left(i\right).$$

Also, the following definition will be needed for our algorithm. We denote by $G^r$ the $r$-th power of $G$, in which $v_1, v_2 \in V$ are adjacent if and only if $\mathrm{dist}_G\left(v, v'\right) \leq r$; and by $G_{|U}$ the sub-graph of $G$ induced by $U \subseteq V$.

**Definition 3.** Let $G = \langle V, E\rangle$ be an undirected connected graph and let $W \subseteq U \subseteq V$. $W$ is called an *$r$-independent set* of $G$, if it is an independent set of $G^r$. Namely,

$$\forall w, w' \in W : \mathrm{dist}_G\left(w, w'\right) \geq r + 1.$$

If $W$ is also a maximal independent set of $(G^r)_{|U}$, it is called a *maximal $r$-independent subset* ($r$-MIS) of $U$. Namely, there is no $r$-independent set $W' \subseteq U$ such that $W \subset W'$.

## 3 Center-based cooperative multi-armed bandits

We now present the center-based policy for the cooperative multi-armed bandit setting, which will give us the desired low individual regret. In the center-based cooperative MAB, not all the agents behave similarly. We partition the agents to three different types.

*Center agents* are the agents that determine the action distribution for all other agents. They work together with their neighbors to minimize their regret. The neighbors of the center agents in the communication graph, *center-adjacent* agents, always copy the action distribution from their neighboring center, and thus the centers gain more information about their own distribution each step.

Other (not center or center-adjacent) agents are *simple agents*, which simply copy the action distribution from one of the centers. Since they are not center-adjacent, they receive the action distribution with delay, through other agents that copy from the same center.

We artificially partition the graph to connected components, such that each center $c$ has its own component, and all the simple agents in the component of $c$ copy their action distribution from it. To obtain a low individual regret, we require the components to have a relatively small diameter, and the center agents to have a high degree in the communication graph. Namely, center agents have the highest or nearly highest degree in their component.

In more detail, we select a set $C \subseteq V$ of center agents. All center agents $c \in C$ use the exponential-weights algorithm with a learning rate $\eta\left(c\right) = \frac{1}{2}\sqrt{\frac{(\ln K)\min\{|\mathcal{N}(c)|, K\}}{KT}}$. The agent set $V$ is partitioned into disjoint subsets $\{V_c \subseteq V \mid c \in C\}$, such that $\mathcal{N}\left(c\right) \subseteq V_c$ for all $c \in C$, and the sub-graph $G_c \equiv G_{|V_c}$ induced by $V_c$ is connected. Notice that since the components are disjoint, the condition $\mathcal{N}\left(c\right) \subseteq V_c$ implies $C$ is a 2-independent set. For all non-centers $v \in V \setminus C$, we denote by $\mathcal{C}\left(v\right) \in C$ the center agent such that $v \in V_{\mathcal{C}(v)}$, and call it the *center of $v$*. All non-center agents $v \in V \setminus C$ copy their distribution from their *origin neighbor $U\left(v\right)$*, which is their neighbor in $G_{\mathcal{C}(v)}$ closest to $\mathcal{C}\left(v\right)$, breaking ties arbitrarily. Namely,

$$U\left(v\right) = \underset{v' \in \mathcal{N}(v) \cap V_{\mathcal{C}(v)}}{\arg\min}\ \mathrm{dist}_{G_{\mathcal{C}(v)}}\left(v', \mathcal{C}\left(v\right)\right).$$

Thus, agent $v$ receives its center's distribution with a delay of $d\left(v\right) = \mathrm{dist}_{G_{\mathcal{C}(v)}}\left(v, \mathcal{C}\left(v\right)\right)$ steps, so for all $t \geq d\left(v\right) + 1$:

$$\boldsymbol{p}_t^v = \boldsymbol{p}_{t-d(v)}^{\mathcal{C}(v)}.$$

Notice that if $v \in \mathcal{N}\left(c\right)$, then $v$ is center-adjacent and it holds $U\left(v\right) = \mathcal{C}\left(v\right)$ and $d\left(v\right) = 1$. For completeness, we define $U\left(c\right) = \mathcal{C}\left(c\right) = c$ and $d\left(c\right) = 0$

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

# Supplementary Material

## A Proofs from Subsection <span style="color:red">5.1</span>

We first bound the regret of the center agents:

**Lemma 15.** *Let $T \geq K^2 \ln K$. Using the center-based policy, the expected regret of each center $c \in C$ satisfies*

$$R_T(c) \leq 4\sqrt{(\ln K)\frac{K}{M(c)}T}.$$

*Proof.* Since $T \geq K^2 \ln K$, we have $\eta(c) = \frac{1}{2}\sqrt{\frac{(\ln K)M(c)}{KT}} \leq \frac{1}{2}\sqrt{\frac{M(c)}{K^3}} \leq \frac{1}{2K}$. Hence, from Lemma 2, we get for any $v \in \mathcal{N}(c) \setminus c$ and $i \in A$:

$$p_t^v(i) = p_{t-1}^c(i) \geq \frac{1}{2}p_t^c(i).$$

Hence,

$$\mathbb{E}_t[B_t^c(i)] = 1 - \prod_{v\in\mathcal{N}(c)}(1 - p_t^v(i))$$

$$\geq 1 - \left(1 - \frac{1}{2}p_t^c(i)\right)^{|\mathcal{N}(c)|}$$

$$\geq 1 - \exp\left(-\frac{1}{2}p_t^c(i)|\mathcal{N}(c)|\right) \qquad\qquad (1 - x \leq e^{-x})$$

$$\geq 1 - \exp\left(-\min\left\{\frac{1}{2}|\mathcal{N}(c)|\,p_t^c(i),1\right\}\right)$$

$$\geq (1 - e^{-1})\min\left\{\frac{1}{2}|\mathcal{N}(c)|\,p_t^c(i),1\right\}, \qquad ((1 - e^{-1})\,x \leq 1 - e^{-x} \text{ for } 0 \leq x \leq 1)$$

and thus,

$$\mathbb{E}_t\left[\hat{\ell}_t^c(i)^2\right] = \mathbb{E}_t\left[\frac{\ell_t(i)^2}{\mathbb{E}_t[B_t^c(i)]^2}B_t^c(i)\right]$$

$$\leq \frac{1}{\mathbb{E}_t[B_t^c(i)]} \qquad\qquad (\ell_t(i) \leq 1)$$

$$\leq \frac{1}{(1 - e^{-1})\min\left\{\frac{1}{2}|\mathcal{N}(c)|\,p_t^c(i),1\right\}}$$

$$\leq 2 + \frac{4}{|\mathcal{N}(c)|\,p_t^c(i)}.$$

By Lemma 1, we now obtain

$$R_T(c) \leq \frac{\ln K}{\eta(c)} + \frac{\eta(c)}{2}\mathbb{E}\left[\sum_{t=1}^{T}\sum_{i=1}^{K}p_t^c(i)\,\mathbb{E}_t\left[\hat{\ell}_t^c(i)^2\right]\right]$$

$$\leq \frac{\ln K}{\eta(c)} + \frac{\eta(c)}{2}\left(2 + 4\frac{K}{|\mathcal{N}(c)|}\right)T$$

$$\leq \frac{\ln K}{\eta(c)} + 4\eta(c)\frac{K}{M(c)}T$$

$$= 4\sqrt{(\ln K)\frac{K}{M(c)}T} \qquad\qquad (\eta(c) = \frac{1}{2}\sqrt{\frac{(\ln K)\,M(c)}{KT}})$$

as claimed. $\qquad\qquad\qquad\qquad\qquad\qquad\qquad\qquad\qquad\qquad\qquad\qquad\qquad\qquad\square$

Since non-center agents use the same distribution as some center, only with delay, we can use this result together with Lemma 2 to bound the regret of all agents in the graph.

**Proof of Theorem 5**

**Theorem 5.** *Let $T \geq K^2 \ln K$. Using the center-based policy, the regret of each agent $v \in V$ satisfies*

$$R_T(v) \leq 7\sqrt{(\ln K)\frac{K}{M(v)}T}.$$

*Proof.* Again, since $T \geq K^2 \ln K$ we have $\eta(v) \leq \frac{1}{2K}$. Recall that $\boldsymbol{p}_t^v = \boldsymbol{p}_{t-d(v)}^{\mathcal{C}(v)}$. Thus, we can use Lemma 2 iteratively to obtain for all $t > d(v)$:

$$
\begin{aligned}
p_t^v(i) &= p_{t-d(v)}^{\mathcal{C}(v)}(i) \\
&\leq p_{t-d(v)+1}^{\mathcal{C}(v)}(i) + \eta(\mathcal{C}(v)) p_{t-d(v)}^{\mathcal{C}(v)}(i) \hat{\ell}_{t-d(v)}^{\mathcal{C}(v)}(i) \\
&\leq \cdots \leq p_t^{\mathcal{C}(v)}(i) + \eta(\mathcal{C}(v)) \sum_{s=1}^{d(v)} p_{t-s}^{\mathcal{C}(v)}(i) \hat{\ell}_{t-s}^{\mathcal{C}(v)}(i),
\end{aligned}
$$

which yields

$$
\begin{aligned}
R_T(v) &= \mathbb{E}\left[\sum_{t=1}^{T} \ell_t(I_t(v)) - \min_{i \in A} \sum_{t=1}^{T} \ell_t(i)\right] \\
&= \mathbb{E}\left[\sum_{t=1}^{T}\sum_{i=1}^{K} p_t^v(i)\ell_t(i) - \min_{i \in A} \sum_{t=1}^{T} \ell_t(i)\right] \\
&\leq d(v) + \mathbb{E}\left[\sum_{t=d(v)}^{T}\sum_{i=1}^{K} p_t^{\mathcal{C}(v)}(i)\ell_t(i) - \min_{i \in A} \sum_{t=d(v)}^{T} \ell_t(i)\right] \\
&\quad + \eta(\mathcal{C}(v))\mathbb{E}\left[\sum_{t=d(v)}^{T}\sum_{i=1}^{K}\sum_{s=1}^{d(v)} p_{t-s}^{\mathcal{C}(v)}(i) \hat{\ell}_{t-s}^{\mathcal{C}(v)}(i)\ell_t(i)\right] \\
&\leq R_T(\mathcal{C}(v)) + d(v) + d(v)\eta(\mathcal{C}(v))T,
\end{aligned}
$$

where the last inequality is implied from

$$\mathbb{E}_{t-s}\left[\sum_{i=1}^{K} p_{t-s}^{\mathcal{C}(v)}(i) \hat{\ell}_{t-s}^{\mathcal{C}(v)}(i)\ell_t(i)\right] \leq \sum_{i=1}^{K} p_{t-s}^{\mathcal{C}(v)}(i)\ell_{t-s}(i) \leq 1.$$

Hence, using Lemma 15 we get

$$
\begin{aligned}
R_T(v) &\leq \left(4\sqrt{\frac{K}{M(\mathcal{C}(v))}} + d(v) + \frac{d(v)}{2}\sqrt{\frac{M(\mathcal{C}(v))}{K}}\right)\sqrt{(\ln K)T} \\
&\leq \left(4\sqrt{\frac{K}{M(\mathcal{C}(v))}} + d(v)\sqrt{\frac{M(\mathcal{C}(v))}{K}}\right)\sqrt{(\ln K)T} \\
&\leq 7\sqrt{(\ln K)\frac{K}{M(v)}T},
\end{aligned}
$$

concluding our proof. $\qquad\square$

## B Proofs from Subsection 5.2

We first show two helpful lemmas that analyze the results of Centers-to-Components. In the following, we denote $\Theta_K = \lfloor 12 \ln K \rfloor$ and $\tau^v = \min_{c \in C} \text{dist}_G(v, c)$ for any $v \in V$.

**Lemma 16.** *Let $\mathcal{C}_t(v), U_t(v), M_t(v)$ be the variables of agent $v$ at iteration $t$ from Centers-to-Components. Then the following properties hold for all $1 \leq t \leq \Theta_K + 1$ and $v \in V \setminus C$:*

1. *$M_t(v) \geq M_{t-1}(v)$.*

2. *If $M_t(v) \neq M_{t-1}(v)$, then $\mathcal{C}_t(v) \neq \mathrm{nil}$ and*
$$M_t(v) = e^{-\frac{1}{6}t} M(\mathcal{C}_t(v)).$$
   *Moreover, $t \geq \mathrm{dist}_G(v, \mathcal{C}_t(v))$.*

3. *If $\tau^v \leq \Theta_K + 1$, then $M_{\tau^v}(v) \geq e^{-\frac{1}{6}\tau^v}$.*

4. *If $\tau^v \leq 6 \ln K$, then $\mathcal{C}_{\Theta_K+1}(v) = \mathcal{C}_{\Theta_K}(v), M_{\Theta_K+1}(v) = M_{\Theta_K}(v)$.*

*Proof.*

1. For center-adjacent agents this is immediate from the algorithm. Otherwise, let $v$ be a simple agent. We proceed by induction over $t$. If $t = 1$, we have $M_t(v) = M_1(v) = M_0(v) = 0$. Assume for all $t > 1$ and $v' \in V \setminus C$ that $M_{t-1}(v') \geq M_{t-2}(v')$. Since we choose $U_t(v)$ to be the neighbor with maximal mass at iteration $t-1$, for any $t > 1$ we get
$$M_t(v) = e^{-\frac{1}{6}} M_{t-1}(U_t(v))$$
$$\geq e^{-\frac{1}{6}} M_{t-1}(U_{t-1}(v))$$
$$\geq e^{-\frac{1}{6}} M_{t-2}(U_{t-1}(v))$$
$$= M_{t-1}(v)$$
   as desired.

2. Again we proceed by induction on $t$. If $t = 1$ and $M_t(v) = M_1(v) \neq M_0(v) = 0$, then $M_0(U_1(v)) \neq 0$, and thus $U_1(v) = \mathcal{C}_1(v) \in C$. Hence, $M_t(v) = M_1(v) = e^{-\frac{1}{6}t} M(\mathcal{C}_1(v))$. For any $t > 1$, we assume the property is true for any $v' \in V \setminus C$ at iteration $t-1$. If $M_t(v) \neq M_{t-1}(v)$ we obtain from property 1 that $M_t(v) > M_{t-1}(v)$, and thus
$$e^{-\frac{1}{6}} M_{t-1}(U_t(v)) > e^{-\frac{1}{6}} M_{t-2}(U_{t-1}(v)).$$
   From the way $U_{t-1}(v)$ is chosen we get
$$M_{t-1}(U_t(v)) > M_{t-2}(U_{t-1}(v)) \geq M_{t-2}(U_t(v)).$$
   Hence, $M_{t-1}(U_t(v)) \neq M_{t-2}(U_t(v))$, and from our assumption
$$M_t(v) = e^{-\frac{1}{6}} M_{t-1}(U_t(v)) = e^{-\frac{1}{6}-\frac{1}{6}(t-1)} M(\mathcal{C}_{t-1}(U_t(v))) = e^{-\frac{1}{6}t} M(\mathcal{C}_t(v)),$$
   where in the last equality we used the fact that $\mathcal{C}_t(v) = \mathcal{C}_{t-1}(U_t(v))$. We also get
$$t - 1 \geq \mathrm{dist}_G(U_t(v), \mathcal{C}_{t-1}(U_t(v))) = \mathrm{dist}_G(U_t(v), \mathcal{C}_t(v)).$$
   Hence, since $\mathrm{dist}_G(v, \mathcal{C}_t(v)) \leq \mathrm{dist}_G(U_t(v), \mathcal{C}_t(v)) + 1$, we obtain $t \geq \mathrm{dist}_G(v, \mathcal{C}_t(v))$ as desired.

3. We proceed by induction on $\tau^v$. If $\tau^v = \min_{c \in C} \mathrm{dist}_G(v, c) = 1$, then $v$ is center-adjacent and thus $\mathcal{C}_1(v) = \arg\min_{c \in C} \mathrm{dist}_G(v, c)$, which gives
$$M_{\tau^v}(v) = M_1(v) = e^{-\frac{1}{6}} M(\mathcal{C}_1(v)) \geq e^{-\frac{1}{6}\tau^v}.$$
   Otherwise, let $v \in V \setminus C$ be a simple agent with $\tau^v > 1$ and $c = \arg\min_{c' \in C} \mathrm{dist}_G(v, c')$. It must have a neighbor $v' \in \mathcal{N}(v)$ such that $c = \arg\min_{c' \in C} \mathrm{dist}_G(v', c')$ and $\tau^{v'} = \mathrm{dist}_G(v', c) = \mathrm{dist}_G(v, c) - 1 = \tau^v - 1$. We assume the property is true for $v'$. From the way the origin neighbor at iteration $\tau^v$ is chosen we obtain that
$$M_{\tau^v}(v) = e^{-\frac{1}{6}} M_{\tau^v-1}(U_{\tau^v}(v))$$
$$\geq e^{-\frac{1}{6}} M_{\tau^v-1}(v')$$
$$= e^{-\frac{1}{6}} M_{\tau^{v'}}(v')$$
$$\geq e^{-\frac{1}{6}\left(1+\tau^{v'}\right)}$$
$$= e^{-\frac{1}{6}\tau^v},$$

as desired.

4. Assuming to the contrary $M_{\Theta_K+1}(v) \neq M_{\Theta_K}(v)$ (or $\mathcal{C}_{\Theta_K+1}(v) \neq \mathcal{C}_{\Theta_K}(v)$), we get

$$
\begin{aligned}
M_{\Theta_K+1}(v) &= e^{-\frac{1}{6}(\Theta_K+1)} M\left(\mathcal{C}_{\Theta_K+1}(v)\right) && \text{(property 2)} \\
&< \frac{1}{K} \\
&\leq e^{-\frac{1}{6}\tau^v} && (\tau^v \leq 6\ln K) \\
&\leq M_{\tau^v}(v) && \text{(property 3)} \\
&\leq M_{\lceil 6\ln K\rceil}(v), && \text{(property 1)}
\end{aligned}
$$

contradicting property 1 and concluding our proof.

$\square$

**Lemma 17.** *Let $\mathcal{C}(v), U(v), M(v)$ be the results of Centers-to-Components. Then the following properties hold for all simple agents $v \in V \setminus C$ such that $2 \leq \min_{c \in C}\operatorname{dist}_G(v,c) \leq 6\ln K - 1$:*

1. *$\mathcal{C}(v) \neq$ nil and $U(v) \neq$ nil.*

2. *$M(v) = e^{-\frac{1}{6}} M(U(v))$.*

3. *$\mathcal{C}(v) = \mathcal{C}(U(v))$.*

*Proof.* Let $v \in V$ be a simple agent such that $\tau^v = \min_{c \in C}\operatorname{dist}_G(v,c) \leq 6\ln K - 1$.

1. We have
$$
\begin{aligned}
M(v) &= M_{\Theta_K+1}(v) \\
&\geq M_{\tau^v}(v) && \text{(property 1 of Lemma 16)} \\
&> 0, && \text{(property 3 of Lemma 16)}
\end{aligned}
$$
and thus it follows from the algorithm that $\mathcal{C}(v) \neq$ nil and $U(v) \neq$ nil as desired.

2. Since $\tau^{U(v)} \leq \tau^v + 1 \leq 6\ln K$, we get:
$$
\begin{aligned}
M(v) &= M_{\Theta_K+1}(v) \\
&= e^{-\frac{1}{6}} M_{\Theta_K}(U(v)) && \text{(from the algorithm)} \\
&= e^{-\frac{1}{6}} M_{\Theta_K+1}(U(v)) && \text{(property 4 of Lemma 16)} \\
&= e^{-\frac{1}{6}} M(U(v)).
\end{aligned}
$$

3. Using property 4 of Lemma 16 again, we obtain
$$
\mathcal{C}(v) = \mathcal{C}_{\Theta_K}(U(v)) = \mathcal{C}_{\Theta_K+1}(U(v)) = \mathcal{C}(U(v)).
$$

$\square$

The next lemma shows that all simple agents choose the best possible agent as their origin neighbor.

**Lemma 18.** *Let $U(v)$ and $M(v)$ be the results of Centers-to-Components. Then for all simple agents $v \in V \setminus C$ such that $2 \leq \min_{c \in C}\operatorname{dist}_G(v,c) \leq 6\ln K - 1$:*
$$
U(v) = \arg\max_{v' \in \mathcal{N}(v)} M(v')
$$

*Proof.* Simple agents choose their origin neighbor to be the one with maximal mass at iteration $\Theta_K = \lfloor 12\ln K\rfloor$. In addition, since $\tau^v = \min_{c \in C}\operatorname{dist}_G(v,c) \leq 6\ln K - 1$ we obtain $\tau^{U(v)} \leq \tau^v + 1 \leq 6\ln K$, so we can use property 4 of Lemma 16 and get:
$$
M(U(v)) = M_{\Theta_K+1}(U(v)) = M_{\Theta_K}(U(v)) \geq M_{\Theta_K}(v') = M_{\Theta_K+1}(v') = M(v')
$$
as desired.

$\square$

**Proof of Lemma 6**

**Lemma 6.** *Let $C \subseteq V$ be a center set that is 2-independent, such that every $v \in V$ holds $\min_{c \in C} \operatorname{dist}_G (v, c) \le 6 \ln K - 1$. Let $\mathcal{C}(v), U(v), M(v)$ be the results of Centers-to-Components. For each $c \in C$, let $V_c$ be its corresponding component, namely, $V_c = \{v \in V \mid \mathcal{C}(v) = c\}$. Then the following properties are satisfied:*

1. *$\{V_c \mid c \in C\}$ are pairwise disjoint and $V = \bigcup_{c \in C} V_c$.*

2. *$\mathcal{N}(c) \subseteq V_c$ and $G_c$ is connected for all $c \in C$.*

3. *$M(v) = e^{-\frac{1}{6} d(v)} M(\mathcal{C}(v))$ and $U(v) = \arg\min_{v' \in \mathcal{N}(v) \cap V_{\mathcal{C}(v)}} d(v')$ for all $v \in V \setminus C$.*

*Proof.*

1. The components are trivially disjoint from the way we defined them. Since for any $v \in V$ we assume $\tau^v = \min_{c \in C} \operatorname{dist}_G(v, c) \le 6 \ln K - 1$, we obtain from property 1 of Lemma 17 that $\mathcal{C}(v) \ne \text{nil}$ and $v \in \bigcup_{c \in C} V_c$ as desired.

2. Since $C$ is 2-independent, it directly follows from the algorithm that $\mathcal{N}(c) \subseteq V_c$ for all $c \in C$. Now, let $v \in V$. For a path of connected agents $v = u_0, \ldots, u_m$ such that $u_{i+1} = U(u_i)$ for any $i < m$, we get from property 3 of Lemma 17 that $\mathcal{C}(u_i) = \mathcal{C}(v)$ for all $i$. From property 2 of Lemma 17 we also obtain that $M(u_i) < M(u_{i+1})$ for all $i < m$ such that $u_i \notin C$, and thus all non-center agents on the path must be different. Hence, if $m \ge N$ we obtain that there must be a center $u$ on the path, and since $u = \mathcal{C}(u) = \mathcal{C}(v)$, we get that $\mathcal{C}(v)$ must be connected to $v$. We obtain that all agents are connected to their center, and thus $G_c$ is connected for all $c \in C$ as claimed.

3. We proceed by induction on $d(v) = \operatorname{dist}_{G_{\mathcal{C}(v)}}(v, \mathcal{C}(v))$. If $d(v) = 1$ (i.e., $v$ is center-adjacent), the statement trivially follows from the algorithm. Otherwise, we assume the statement is true for all $v' \in V \setminus C$ such that $d(v') < d(v)$. Since $G_{\mathcal{C}(v)}$ is connected from property 2, there must be some $v' \in \mathcal{N}(v) \cap V_{\mathcal{C}(v)}$ such that $d(v) = d(v') + 1$, and thus we get from the induction assumption that $M(v') = e^{-\frac{1}{6} d(v')} M(\mathcal{C}(v))$. From Lemma 18, we get that $M(U(v)) \ge M(v')$, and using property 2 of Lemma 17 we obtain

$$M(v) \ge e^{-\frac{1}{6}} M(v') = e^{-\frac{1}{6}\left(d(v') + 1\right)} M(\mathcal{C}(v)) = e^{-\frac{1}{6} d(v)} M(\mathcal{C}(v)). \qquad (1)$$

As before, from Lemma 17 there is a path $v = u_0, \ldots, u_m = \mathcal{C}(v)$ from $v$ to its center such that $U(u_i) = u_{i+1}$ for any $i < m$ and $\mathcal{C}(u_i) = \mathcal{C}(v)$ for all $i$. We must have $m \ge \operatorname{dist}_{G_{\mathcal{C}(v)}}(v, \mathcal{C}(v)) = d(v)$, and using property 2 of Lemma 17 iteratively we get

$$M(v) = e^{-\frac{1}{6}} M(u_1) = \cdots = e^{-\frac{1}{6} m} M(\mathcal{C}(v)) \le e^{-\frac{1}{6} d(v)} M(\mathcal{C}(v)).$$

Combining with Eq. (1) we get $M(v) = e^{-\frac{1}{6} d(v)} M(\mathcal{C}(v))$ as desired. From property 3 of Lemma 17 we have $U(v) \in \mathcal{N}(v) \cap V_{\mathcal{C}(v)}$, and using Lemma 18 we get

$$
\begin{aligned}
U(v) &= \arg\max_{v' \in \mathcal{N}(v) \cap V_{\mathcal{C}(v)}} M(v') \\
&= \arg\max_{v' \in \mathcal{N}(v) \cap V_{\mathcal{C}(v)}} e^{-\frac{1}{6} d(v')} M(\mathcal{C}(v)) \\
&= \arg\min_{v' \in \mathcal{N}(v) \cap V_{\mathcal{C}(v)}} d(v'),
\end{aligned}
$$

concluding our proof.

$\square$

# C Proofs from Subsection 5.3

**Proof of Lemma 7**

**Lemma 7.** *Let $C \subseteq V$ be the center set returned by Compute-Centers-Informed. Then:*

1. *$C$ is 2-independent.*

2. *For all $v \in V$, $\min_{c \in C} \text{dist}_G(v, c) \leq 6 \ln K - 1$.*

*Proof.*

1. The statement follows directly from the fact the agent $v$ that is added to the center set at iteration $t$ holds $\min_{c \in C_t} \text{dist}_G(v, c) \geq 3$.

2. When the algorithm terminates there are no unsatisfied agents. Hence, for all $v \in V$, either $\min_{c \in C} \text{dist}_G(v, c) \leq 2$, in which case we are done, or $M(v) \geq \min\{|\mathcal{N}(v)|, K\} \geq 2$. In the latter case we obtain from properties 1 and 2 of Lemma 16:

$$2 \leq M(v) = \exp\left(-\frac{1}{6}\text{dist}_G(v, \mathcal{C}(v))\right) M(\mathcal{C}(v)) \leq \exp\left(-\frac{1}{6}\text{dist}_G(v, \mathcal{C}(v))\right) K,$$

and thus $\min_{c \in C} \text{dist}_G(v, c) \leq 6 \ln K - 1$ as desired.

$\square$

**Proof of Theorem 8**

**Theorem 8.** *Let $C \subseteq V$ be the center set returned by Compute-Centers-Informed, and let $\{V_c \subseteq V \mid c \in C\}$ be the components resulted from Centers-to-Components. For every $v \in V$:*

$$M(v) \geq e^{-1} \min\{|\mathcal{N}(v)|, K\}.$$

*Proof.* For any center $v \in C$ this is trivial. Since all agents are satisfied when the algorithm terminates, each $v \in V \setminus C$ must either hold $M(v) \geq \min\{|\mathcal{N}(v)|, K\}$ or $\min_{c \in C} \text{dist}_G(v, c) \leq 2$. Hence, we only need to prove the claim for each non-center agent $v \in V \setminus C$ in distance at most 2 from the center set.

We first inspect the case that the agent is not center-adjacent, namely, $\min_{c \in C} \text{dist}_G(v, c) = 2$. Let $t_0$ be the last iteration such that $\min_{c \in C_{t_0}} \text{dist}_G(v, c) \geq 3$. Note that this means $\text{dist}_G(v, c_{t_0}) = 2$. In the case that $v \notin S_{t_0}$, $v$ is satisfied, and since $\min_{c \in C_{t_0}} \text{dist}_G(v, c) \geq 3$, it must hold $M_{t_0}(v) \geq \min\{|\mathcal{N}(v)|, K\}$. Also, $c_{t_0} \in S_{t_0}$ and thus $3 \leq \min_{c \in C_{t_0}} \text{dist}_G(c_{t_0}, c)$ and $M_{t_0}(c_{t_0}) < \min\{|\mathcal{N}(c_{t_0})|, K\}$. Now, property 2 of Lemma 16 gives

$$\exp\left(-\frac{1}{6}\min_{c \in C_{t_0}} \text{dist}_G(v, c)\right) K \geq M_{t_0}(v) \geq \min\{|\mathcal{N}(v)|, K\}.$$

Recall that $v \in \mathcal{N}(v)$, so $|\mathcal{N}(v)| \geq 2$ and thus $\exp\left(-\frac{1}{6}\min_{c \in C_{t_0}} \text{dist}_G(v, c)\right) K \geq 2$. Hence, $3 \leq \min_{c \in C_{t_0}} \text{dist}_G(v, c) \leq 6 \ln K - 6 \ln 2 \leq 6 \ln K - 4$ and thus $3 \leq \min_{c \in C_{t_0}} \text{dist}_G(c_{t_0}, c) \leq 6 \ln K - 2$. Let $u$ be an agent that is a common neighbor of $v$ and $c_{t_0}$, namely, $u \in \mathcal{N}(v) \cap \mathcal{N}(c_{t_0})$. We obtain $2 \leq \min_{c \in C_{t_0}} \text{dist}_G(u, c) \leq 6 \ln K - 3$ as well. We can now use Lemma 18 on $c_{t_0}$ and $u$ to obtain

$$\min\{|\mathcal{N}(c_{t_0})|, K\} > M_{t_0}(c_{t_0}) \geq e^{-\frac{1}{6}} M_{t_0}(u) \geq e^{-\frac{2}{6}} M_{t_0}(v) \geq e^{-\frac{2}{6}} \min\{|\mathcal{N}(v)|, K\}.$$

In the other case that $v \in S_{t_0}$, since $c_{t_0} = \arg\max_{v' \in S_{t_0}} |\mathcal{N}(v')|$, we obtain $|\mathcal{N}(v)| \leq |\mathcal{N}(c_{t_0})|$, and anyway $\min\{|\mathcal{N}(c_{t_0})|, K\} \geq e^{-\frac{2}{6}} \min\{|\mathcal{N}(v)|, K\}$. In all further iterations $t > t_0$, we can use Lemma 18 on $v$ to obtain

$$M_t(v) \geq e^{-\frac{1}{6}} M_t(u) = e^{-\frac{2}{6}} \min\{|\mathcal{N}(c_{t_0})|, K\} \geq e^{-\frac{4}{6}} \min\{|\mathcal{N}(v)|, K\},$$

as desired.

Now we look at the case where $v$ is center-adjacent and $\min_{c \in C} \operatorname{dist}_G (v, c) = 1$. Again, let $t_0$ be the last iteration such that $\min_{c \in C_{t_0}} \operatorname{dist}_G (v, c) \geq 2$, and thus $\operatorname{dist}_G (v, c_{t_0}) = 1$. In the case that $v \notin S_{t_0}$, either $M_{t_0} (v) \geq \min \{|\mathcal{N} (v)|, K\}$ or $\min_{c \in C_{t_0}} \operatorname{dist}_G (v, c) = 2$, in which case we obtain from before that $M_{t_0} (v) \geq e^{-\frac{4}{6}} \min \{|\mathcal{N} (v)|, K\}$. As before, we can use Lemma 18 on $c_{t_0}$ and get

$$\min \{|\mathcal{N} (c_{t_0})|, K\} > M_{t_0} (c_{t_0}) \geq e^{-\frac{1}{6}} M_{t_0} (v) \geq e^{-\frac{5}{6}} \min \{|\mathcal{N} (v)|, K\}.$$

In the other case that $v \in S_{t_0}$, again we obtain $\min \{|\mathcal{N} (c_{t_0})|, K\} \geq e^{-\frac{5}{6}} \min \{|\mathcal{N} (v)|, K\}$. In all further iterations $t > t_0$, we get

$$M_t (v) = e^{-\frac{1}{6}} \min \{|\mathcal{N} (c_{t_0})|, K\} \geq e^{-1} \min \{|\mathcal{N} (v)|, K\},$$

concluding our proof. $\qquad\square$

## D  Proofs from Subsection 5.4

We first present the next lemma, which will help us with the analysis of Compute-Centers-Uninformed. In the following, we denote $\Delta^v = K - \min \{|\mathcal{N} (v)|, K\}$.

**Lemma 19.** *Let $C \subseteq V$ be the center set returned by Compute-Centers-Uninformed, and let $\{V_c \subseteq V \mid c \in C\}$ be the components resulted from Centers-to-Components. For any $v \in V$ such that $v \notin S_{\Delta^v}$, either $M (v) \geq \min \{|\mathcal{N} (v)|, K\}$, or there is some $c \in C$ such that $|\mathcal{N} (c)| \geq e^{-\frac{1}{6}} |\mathcal{N} (v)|$ and $\operatorname{dist}_G (v, c) \leq 2$.*

*Proof.* Let $v \in V$ be an agent such that $v \notin S_{\Delta^v}$. At iteration $\Delta^v - 1$, it follows directly from the algorithm that either $\min_{c \in C_{\Delta^v - 1}} \operatorname{dist}_G (v, c) \leq 2$ or $M_{\Delta^v - 1} (v) \geq \min \{|\mathcal{N} (v)|, K\}$. In the first case, since $|\mathcal{N} (v)| \leq |\mathcal{N} (c)|$ for all $c \in C_{\Delta^v - 1} \subseteq C$, we are done.

Otherwise, we denote by $c^v = \mathcal{C}_{\Delta^v - 1} (v) \neq \text{nil}$ the center of agent $v$ at iteration $\Delta^v - 1$. Note that from properties 1 and 2 of Lemma 16, we obtain:

$$e^{-\frac{1}{6} \operatorname{dist}_G (v, c^v)} M (c^v) \geq M_{\Delta^v - 1} (v) \geq \min \{|\mathcal{N} (v)|, K\} \geq 2,$$

and thus $\operatorname{dist}_G (v, c^v) \leq 6 \ln K - 1$. From Lemma 17, we get that $\mathcal{C}_{\Delta^v - 1} (U_{\Delta^v - 1} (v)) = c^v$ and

$$\begin{aligned}
e^{-\frac{1}{6} \operatorname{dist}_G (U_{\Delta^v - 1}(v), c^v)} M (c^v) &\geq M_{\Delta^v - 1} (U_{\Delta^v - 1} (v)) \\
&= e^{\frac{1}{6}} M_{\Delta^v - 1} (v) \\
&\geq \min \{|\mathcal{N} (v)|, K\} \\
&\geq 2.
\end{aligned}$$

Thus, we get $\operatorname{dist}_G (U_{\Delta^v - 1} (v), c^v) \leq 6 \ln K - 1$ as well. Using this fact iteratively, we get that there is a path $v = u_0, \ldots, u_m = c^v$ such that $U_{\Delta^v - 1} (u_i) = u_{i+1}$, $\operatorname{dist}_G (u_i, c^v) \leq 6 \ln K - 1$ and $M_{\Delta^v - 1} (u_{i+1}) = e^{\frac{1}{6}} M_{\Delta^v - 1} (u_i)$ for any $i < m$. Notice that this also means $M_{\Delta^v - 1} (v) = e^{-\frac{1}{6} m} M (c^v)$.

Now, assume to the contrary some simple agent on the path other than $v$ becomes a center or center-adjacent after iteration $\Delta^v - 1$, and let $u_j$ be the first such agent, where $1 \leq j < m - 1$. Let $u \in \mathcal{N} (u_j)$ be the neighbor of $u_j$ that joins the center set. Note that since $\Delta^u \geq \Delta^v$, we obtain $|\mathcal{N} (u)| \leq |\mathcal{N} (v)|$. At iteration $\Delta^u - 1$, all agents in the path are still simple agents (except $c^v$ and $u_{m-1}$), so we can use Lemma 18 iteratively to obtain

$$\begin{aligned}
M_{\Delta^u - 1} (u) &\geq e^{-\frac{1}{6}} M_{\Delta^u - 1} (u_j) \\
&\geq \cdots \geq e^{-\frac{1}{6} (m-j)} M (u_{m-1}) \\
&= e^{-\frac{1}{6} (m-j+1)} M (c^v) \\
&\geq e^{-\frac{1}{6} m} M (c^v) \\
&= M_{\Delta^v - 1} (v) \\
&\geq \min \{|\mathcal{N} (v)|, K\} \\
&\geq \min \{|\mathcal{N} (u)|, K\}.
\end{aligned}$$

Hence, $u \notin S_{\Delta^v}$ which gives $u \notin C_{\Delta^u}$, and thus $u_j$ remains a simple agent. We get that all simple agents on the path at iteration $\Delta^v - 1$ except $v$ must remain simple agents when the algorithm terminates. If $v$ remain a simple agent as well, we obtain from Lemma 18 that

$$M(v) \geq e^{-\frac{1}{6}} M(u_1) \geq \cdots \geq e^{-\frac{1}{6}m} M(c^v) = M_{\Delta^v - 1}(v) \geq \min\{|\mathcal{N}(v)|, K\}$$

as desired. We are left with the case that some $u \in \mathcal{N}(v)$ becomes a center after iteration $\Delta^v$, and thus $M_{\Delta^u - 1}(u) < \min\{|\mathcal{N}(u)|, K\}$. We can again use Lemma 18 iteratively to get

$$\begin{aligned}
\min\{|\mathcal{N}(u)|, K\} &> M_{\Delta^u - 1}(u) \\
&\geq e^{-\frac{1}{6}} M_{\Delta^u - 1}(v) \\
&\geq \cdots \geq e^{-\frac{1}{6}m} M(u_{m-1}) \\
&= e^{-\frac{1}{6}(m+1)} M(c^v) \\
&= e^{-\frac{1}{6}} M_{\Delta^v - 1}(v) \\
&\geq e^{-\frac{1}{6}} \min\{|\mathcal{N}(v)|, K\},
\end{aligned}$$

concluding our proof. $\qquad\square$

**Proof of Lemma 10**

**Lemma 10.** *Compute-Centers-Uninformed runs for less than $12K \ln\left(K^2 \bar{N} T\right)$ steps.*

*Proof.* There are $K$ iterations in Compute-Centers-Uninformed, such that at each iteration the agents run Luby's algorithm for $4 \left\lceil 3 \ln\left(\bar{N}\sqrt{KT}\right)\right\rceil$ steps, and Centers-to-Components for $\Theta_K + 1 = \lfloor 12 \ln K \rfloor + 1$ steps. We obtain that Compute-Centers-Uninformed terminates after

$$\left(4 \left\lceil 3 \ln\left(\bar{N}\sqrt{KT}\right)\right\rceil + \lfloor 12 \ln K \rfloor + 1\right) K \leq 12K \ln\left(K^2 \bar{N} T\right)$$

steps. $\qquad\square$

**Proof of Lemma 11**

**Lemma 11.** *Let $C \subseteq V$ be the center set resulted from Compute-Centers-Uninformed, such that Luby's algorithm succeeded at all iterations of the algorithm. Then:*

1. *$C$ is 2-independent.*

2. *For all $v \in V$, $\min_{c \in C} \operatorname{dist}_G(v, c) \leq 6 \ln K - 1$.*

*Proof.*

1. We get that at each iteration, a 2-independent set is added to the center set, such that every agent $v$ in that set holds $\min_{c \in C_{t-1}} \operatorname{dist}_G(v, c) \geq 3$. Hence, the final center set is 2-independent as claimed.

2. From Lemma 19 we have either $\min_{c \in C} \operatorname{dist}_G(v, c) \leq 2$, in which case we are done, or $M(v) \geq \min\{|\mathcal{N}(v)|, K\} \geq 2$. In the latter case we obtain:

$$\begin{aligned}
2 &\leq M(v) \\
&\leq \exp\left(-\frac{1}{6} \operatorname{dist}_G(v, \mathcal{C}(v))\right) M(\mathcal{C}(v)) \qquad \text{(Properties 1 and 2 of Lemma 16)} \\
&\leq \exp\left(-\frac{1}{6} \operatorname{dist}_G(v, \mathcal{C}(v))\right) K,
\end{aligned}$$

and thus $\min_{c \in C} \operatorname{dist}_G(v, c) \leq 6 \ln K - 1$ as desired.

$\qquad\square$

**Proof of Theorem 12**

**Theorem 12.** *Let $C \subseteq V$ be the center set resulted from Compute-Centers-Uninformed, such that Luby's algorithm succeeded at all iterations of the algorithm, and also let $\{V_c \subseteq V \mid c \in C\}$ be the components resulted from Centers-to-Components. For every $v \in V$:*

$$M(v) \geq e^{-1} \min\{|\mathcal{N}(v)|, K\}.$$

*Proof.* In the case that $v \in S_{\Delta^v}$, since $W_{\Delta^v}$ is a maximal 2-independet set of $S_{\Delta^v}$, we get that either $v \in W_{\Delta^v} \subseteq C$ or $\text{dist}_G(v, v') \leq 2$ for some $v' \in W_{\Delta^v} \subseteq C$. In the case that $v \notin S_{\Delta^v}$, we obtain from Lemma 19 that either $M(v) \geq \min\{|\mathcal{N}(v)|, K\}$, in which case we are done, or there is some center $c' \in C$ such that $\text{dist}_G(v, c') \leq 2$ and $e^{-\frac{1}{6}} \min\{|\mathcal{N}(v)|, K\} \leq \min\{|\mathcal{N}(c')|, K\}$.

Hence we only need to prove the theorem for the case that there is some center $c' \in C$ such that $\text{dist}_G(v, c') \leq 2$ and $e^{-\frac{1}{6}} \min\{|\mathcal{N}(v)|, K\} \leq \min\{|\mathcal{N}(c')|, K\}$. We first inspect the case that $v$ is not a center or center-adjacent. Let $u$ be an agent that is a common neighbor of $v$ and $c'$, namely, $u \in \mathcal{N}(v) \cap \mathcal{N}(c')$. Lemma 18 yields

$$M(v) \geq e^{-\frac{1}{6}} M(u) = e^{-\frac{2}{6}} \min\{|\mathcal{N}(c')|, K\} \geq e^{-\frac{3}{6}} \min\{|\mathcal{N}(v)|, K\},$$

as desired. If $v$ is a center the claim is trivial, so we are left with the case that $v$ is center-adjacent to a center $c \in C$. Note that $\text{dist}_G(c, c') \leq 3$. If $\min\{|\mathcal{N}(c')|, K\} \leq \min\{|\mathcal{N}(c)|, K\}$ we are done. Otherwise, in the case that $\min\{|\mathcal{N}(c)|, K\} < \min\{|\mathcal{N}(c')|, K\}$, we obtain

$$
\begin{aligned}
\min\{|\mathcal{N}(c)|, K\} &> M_{\Delta^c - 1}(c) \\
&\geq e^{-\frac{3}{6}} M_{\Delta^c - 1}(c') && \text{(iterative application of Lemma 18)} \\
&= e^{-\frac{3}{6}} \min\{|\mathcal{N}(c')|, K\} && (c' \in C_{\Delta^c - 1}) \\
&\geq e^{-\frac{4}{6}} \min\{|\mathcal{N}(v)|, K\}.
\end{aligned}
$$

Hence,

$$M(v) = e^{-\frac{1}{6}} \min\{|\mathcal{N}(c)|, K\} \geq e^{-\frac{5}{6}} \min\{|\mathcal{N}(v)|, K\},$$

concluding our proof. $\qquad\square$

**Proof of Corollary 13**

**Corollary 13.** *Let $T \geq K^2 \ln K$ and $\bar{N} \geq N$. Let $C \subseteq V$ be the center set resulted from Compute-Centers-Uninformed, and let $\{V_c \subseteq V \mid c \in C\}$ be the components resulted from Centers-to-Components. Using the center-based policy, we obtain for every $v \in V$:*

$$R_T(v) \leq 12\left(K \ln\left(K^2 \bar{N} T\right) + \sqrt{(\ln K)\left(1 + \frac{K}{|\mathcal{N}(v)|}\right)T}\right) + 1 = \tilde{O}\left(\sqrt{\left(1 + \frac{K}{|\mathcal{N}(v)|}\right)T}\right).$$

*Proof.* Luby's algorithm succeeds with probability $1 - \frac{1}{KT}$ at each iteration of Compute-Centers-Uninformed. Hence, from the union bound, it succeeds at all iterations with probability $1 - \frac{1}{T}$. In that case, from Lemma 11, we can use Theorem 5 and Theorem 12 to bound the expected regret of agent $v$ after Compute-Centers-Uninformed finished by:

$$7\sqrt{(\ln K)\frac{K}{M(v)}T} \leq 7\sqrt{(\ln K)\,e\frac{K}{\min\{|\mathcal{N}(v)|, K\}}T} \leq 12\sqrt{(\ln K)\left(1 + \frac{K}{|\mathcal{N}(v)|}\right)T}.$$

From Lemma 10, Compute-Centers-Uninformed finishes after no more than $12K \ln\left(K^2 \bar{N} T\right)$ steps, so the overall expected regret in this case is bounded by:

$$12\left(K \ln\left(K^2 \bar{N} T\right) + \sqrt{(\ln K)\left(1 + \frac{K}{|\mathcal{N}(v)|}\right)T}\right).$$

In the case that Luby's algorithm failed at one of the iterations, we can bound the regret by $T$, the maximal regret possible. Hence, we obtain the desired result:

$$R_T\left(v\right) \leq 12\left(1-\frac{1}{T}\right)\left(K\ln\left(K^2\bar{N}T\right)+\sqrt{\left(\ln K\right)\left(1+\frac{K}{|\mathcal{N}\left(v\right)|}\right)T}\right)+\frac{1}{T}T$$

$$\leq 12\left(K\ln\left(K^2\bar{N}T\right)+\sqrt{\left(\ln K\right)\left(1+\frac{K}{|\mathcal{N}\left(v\right)|}\right)T}\right)+1.$$

$\square$

## E  Proofs from Subsection 5.5

**Proof of Corollary 14**

**Corollary 14.** *Let $T \geq K^2\ln K$. Let $C \subseteq V$ be the center set resulted from Compute-Centers-Informed or Compute-Centers-Uninformed, and let $\{V_c \subseteq V \mid c \in C\}$ be the components resulted from Centers-to-Components. Using the center-based policy, we get:*

$$\frac{1}{N}\sum_{v\in V}R_T\left(v\right)=\widetilde{O}\left(\sqrt{\left(1+\frac{K}{N}\alpha\left(G\right)\right)T}\right).$$

*Proof.* Using either Compute-Centers-Informed or Compute-Centers-Uninformed to partition the graph for the center-based policy, we get from Corollaries 9 and 13 that for all $v \in V$:

$$R_T\left(v\right)=\widetilde{O}\left(\sqrt{\left(1+\frac{K}{|\mathcal{N}\left(v\right)|}\right)T}\right).$$

Hence,

$$\frac{1}{N}\sum_{v\in V}R_T\left(v\right)=\widetilde{O}\left(\frac{1}{N}\sum_{v\in V}\sqrt{\left(1+\frac{K}{|\mathcal{N}\left(v\right)|}\right)T}\right)=\widetilde{O}\left(\frac{1}{\sqrt{N}}\sqrt{\sum_{v\in V}\left(1+\frac{K}{|\mathcal{N}\left(v\right)|}\right)T}\right),$$

where the last equality is due to the Cauchy–Schwarz inequality. Since $\sum_{v\in V}\frac{1}{|\mathcal{N}(v)|}\leq\alpha\left(G\right)$ [Wei, 1981], we obtain:

$$\frac{1}{N}\sum_{v\in V}R_T\left(v\right)=\widetilde{O}\left(\sqrt{\left(1+\frac{K}{N}\sum_{v\in V}\frac{1}{|\mathcal{N}\left(v\right)|}\right)T}\right)=\widetilde{O}\left(\sqrt{\left(1+\frac{K}{N}\alpha\left(G\right)\right)T}\right)$$

as desired. $\square$

## F  Luby's algorithm

Let $G=\langle V,E\rangle$ be an undirected connected graph and let $U \subseteq V$. We can find a 2-MIS of $U$ in a distributed manner with high probability by using Luby's algorithm [Luby, 1986, Alon et al., 1986] on $\left(G^2\right)_{|U}$, detailed in Algorithm 6.

At each iteration of the algorithm, every agent in $U$ picks a number uniformly from $[0,1]$. Agents that picked the maximal number among their neighbors of distance 2 join the 2-MIS, and their neighbors of distance 2 stop participating. A 2-MIS is computed after $T_\delta=\left\lceil 3\ln\left(\frac{|V|}{\sqrt{\delta}}\right)\right\rceil$ iterations with probability $1-\delta$.

To simulate communication over $G^2$, we use 2 steps to deliver a message. First, the agents send their message. Then, the agents send a message based on the messages they received in the previous step. In Luby's algorithm, agents only need to know the agent in their neighborhood with the maximal random number, or whether an agent in their neighborhood joined the MIS. Hence, every message has length of order $\widetilde{O}\left(1\right)$.

---

**Algorithm 6** Luby's algorithm on $\left(G^2\right)_{|U}$ - agent $v$

---

**Parameters:** Agent set $U \subseteq V$; Error probability $\delta > 0$.
**Initialize:** Participating agents $P_0 = U$.
1: $T_\delta = \left\lceil 3\ln\left(\frac{|V|}{\sqrt{\delta}}\right)\right\rceil$
2: **for** $1 \leq t \leq T_\delta$ **do**
3:      **if** $v \in P_t$ **then**
4:          Pick a number $r_t^v$ uniformly from $[0,1]$.
5:          Send the following message to the set $\mathcal{N}(v)$: $m_{t,1}(v) = \langle v, t, 1, r_t^v \rangle$.
6:      **end if**
7:      Receive all messages $m_{t,1}(v')$ from $v' \in \mathcal{N}(v)$.
8:      **if** $\mathcal{N}(v) \cap P_t \neq \emptyset$ **then**
9:          Set $u_t = \arg\max_{v' \in \mathcal{N}(v) \cap P_t}\left(r_t^{v'}\right)$.
10:          Send the following message to the set $\mathcal{N}(v)$: $m_{t,2}(v) = \langle u_t, t, 2, r_t^u \rangle$.
11:      **end if**
12:      Receive all messages $m_{t,2}(v')$ from $v' \in \mathcal{N}(v)$.
13:      **if** $v = \arg\max_{v' \in P_t \wedge \mathrm{dist}_G(v,v') \leq 2}\left(r_t^{v'}\right)$ **then**
14:          **Join the 2-MIS of** $U$.
15:          Send the following message to the set $\mathcal{N}(v)$: $m_{t,3}(v) = \langle v, t, 3, \mathrm{JOINED}\rangle$.
16:      **end if**
17:      Receive all messages $m_{t,3}(v')$ from $v' \in \mathcal{N}(v)$.
18:      **if** $\exists v' \in \mathcal{N}(v)\,(v'$ joined the 2-MIS$)$ **then**
19:          Send the following message to the set $\mathcal{N}(v)$: $m_{t,4}(v) = \langle v, t, 4, \mathrm{NEIGHBOR\text{-}JOINED}\rangle$.
20:      **end if**
21:      Receive all messages $m_{t,4}(v')$ from $v' \in \mathcal{N}(v)$.
22:      **if** $v \in P_t$ and $\exists v' \in P_t\,(\mathrm{dist}_G(v,v') \leq 2 \wedge v'$ joined the 2-MIS$)$ **then**
23:          Stop participating: $v \notin P_{t+1}$.
24:      **else if** $v \in P_t$ **then**
25:          Continue participating: $v \in P_{t+1}$.
26:      **end if**
27: **end for**

---

For completeness we also provide an overview of the analysis. It follows directly from the algorithm that it outputs an independent set of $\left(G^2\right)_{|U}$. We only need to show it is maximal with high probability, and we prove it using the following lemma (for proof, see [Luby, 1986, Alon et al., 1986]):

**Lemma 20.** *Let $P_t \subseteq U$ be the set of participating agents at iteration $t$ of Luby's algorithm on $\left(G^2\right)_{|U}$, and let $m_t$ be the number of edges of $\left(G^2\right)_{|P_t}$. We obtain for all $t \geq 1$:*

$$\mathbb{E}\left[m_{t+1}\right] \leq \frac{1}{2}\mathbb{E}\left[m_t\right].$$

With this lemma we can now show Luby's algorithm indeed outputs a 2-MIS with high probability.

**Corollary 21.** *Let $W \subseteq U$ be the result of Luby's algorithm on $\left(G^2\right)_{|U}$. Then with probability $1 - \frac{1}{\delta}$, $W$ is a 2-MIS of $U$.*

*Proof.* As we previously mentioned, we only need to show $W$ is a maximal independent set with probability $1 - \delta$. This is equivalent to the statement that $P_{T_\delta+1}$ is empty. If we denote the number of edges of $\left(G^2\right)_{|P_t}$ by $m_t$, we get that it suffices to prove that $m_{T_\delta} = 0$ with high probability. By an iterative application of Lemma 20 we obtain:

$$\mathbb{E}\left[m_{T_\delta}\right] \leq \frac{1}{2}\mathbb{E}\left[m_{T_\delta-1}\right] \leq \cdots \leq \frac{1}{2^{T_\delta}}\mathbb{E}\left[m_0\right] \leq \frac{|V|^2}{2^{T_\delta}}.$$

Hence, we can conclude our proof with Markov's inequality:

$$\Pr\left[m_{T_\delta} \neq 0\right] = \Pr\left[m_{T_\delta} \geq 1\right] \leq \mathbb{E}\left[m_{T_\delta}\right] \leq \frac{|V|^2}{2^{T_\delta}} = \frac{|V|^2}{2^{\left\lceil 3\ln\left(\frac{|V|}{\sqrt{\delta}}\right)\right\rceil}} \leq \delta.$$

---

**Algorithm 7** The exponential-weights algorithm (Exp3)

---

**Parameters:** Number of arms $K$; Time horizon $T$; Learning rate $\eta(v)$.
**Initialize:** $w_1^v(i) \leftarrow \frac{1}{K}$ for all $i \in A$.

1: **for** $1 \leq t \leq T$ **do**
2:     Set $p_t^v(i) \leftarrow \frac{w_t^v(i)}{W_t^v}$ for all $i \in A$, where $W_t^v = \sum_{i \in A} w_t^v(i)$.
3:     Play an action $I_t(v)$ drawn from $\boldsymbol{p}_t^v = \langle p_t^v(1), \ldots, p_t^v(K) \rangle$.
4:     Observe loss $\ell_t(I_t(v))$.
5:     Update for all $i \in A$: $w_{t+1}^v(i) \leftarrow w_t^v(i) \exp\left(-\eta(v)\hat{\ell}_t^v(i)\right)$, where

$$\hat{\ell}_t^v(i) = \frac{\ell_t(i)}{\mathbb{E}_t[B_t^v(i)]} B_t^v(i),$$

    and $B_t^v(i)$ it the event that $v$ observed $\ell_t(I_t(v))$.
6: **end for**

---

$\square$

# G  Proofs from Section 2

For completeness, we give proofs for the preliminary lemmas. The exponential-weights algorithm is given in Algorithm 7.

**Proof of Lemma 1**

**Lemma 1.** *Assuming agent $v$ uses the exponential-weights algorithm, its expected regret satisfies*

$$R_T(v) \leq \frac{\ln K}{\eta(v)} + \frac{\eta(v)}{2} \mathbb{E}\left[\sum_{t=1}^{T} \sum_{i=1}^{K} p_t^v(i) \hat{\ell}_t^v(i)^2\right].$$

*Proof.* We have

$$
\begin{aligned}
\frac{W_{t+1}^v}{W_t^v} &= \sum_{i \in A} \frac{w_{t+1}^v(i)}{W_t^v} \\
&= \sum_{i \in A} \frac{w_t^v(i)}{W_t^v} \exp\left(-\eta(v)\hat{\ell}_t^v(i)\right) \\
&= \sum_{i \in A} p_t^v(i) \exp\left(-\eta(v)\hat{\ell}_t^v(i)\right) \\
&\leq \sum_{i \in A} p_t^v(i) \left(1 - \eta(v)\hat{\ell}_t^v(i) + \eta(v)^2 \hat{\ell}_t^v(i)^2\right) \qquad (e^{-x} \leq 1 - x + \frac{1}{2}x^2 \text{ for } x \geq 0) \\
&= 1 - \eta(v) \sum_{i \in A} p_t^v(i) \hat{\ell}_t^v(i) + \frac{\eta(v)^2}{2} \sum_{i \in A} p_t^v(i) \hat{\ell}_t^v(i)^2.
\end{aligned}
$$

Taking logs and using $\ln(1 + x) \leq x$ we obtain

$$\ln \frac{W_{t+1}^v}{W_t^v} \leq -\eta(v) \sum_{i \in A} p_t^v(i) \hat{\ell}_t^v(i) + \frac{\eta(v)^2}{2} \sum_{i \in A} p_t^v(i) \hat{\ell}_t^v(i)^2.$$

Summing gives

$$\ln W_{T+1}^v \leq -\eta(v) \sum_{t=1}^{T} \sum_{i \in A} p_t^v(i) \hat{\ell}_t^v(i) + \frac{\eta(v)^2}{2} \sum_{t=1}^{T} \sum_{i \in A} p_t^v(i) \hat{\ell}_t^v(i)^2. \tag{2}$$

Now, for any fixed action $k$ we also have

$$\ln W_{T+1}^v \geq \ln w_{T+1}^v (k) = -\eta (v) \sum_{t=1}^{T} \hat{\ell}_t^v (k) - \ln K.$$

Combining with Eq. (2) we obtain

$$\sum_{t=1}^{T} \sum_{i \in A} p_t^v (i) \, \hat{\ell}_t^v (i) - \sum_{t=1}^{T} \hat{\ell}_t^v (k) \leq \frac{\ln K}{\eta (v)} + \frac{\eta (v)^2}{2} \sum_{t=1}^{T} \sum_{i \in A} p_t^v (i) \, \hat{\ell}_t^v (i)^2 .$$

This is true for every $k \in A$. Note that $\mathbb{E}[\cdot] = \mathbb{E}[\mathbb{E}_t[\cdot]]$, and since $\mathbb{E}_t[\ell_t(I_t(v))] = \sum_{i \in A} p_t^v(i) \ell_t(i)$ and $\mathbb{E}_t\left[\hat{\ell}_t^v(i)\right] = \ell_t(i)$, we get

$$\begin{aligned}
R_T(v) &= \mathbb{E}\left[\sum_{t=1}^{T} \ell_t(I_t(v)) - \min_{i \in A} \sum_{t=1}^{T} \ell_t(i)\right] \\
&\leq \mathbb{E}\left[\sum_{t=1}^{T} \ell_t(I_t(v))\right] - \min_{i \in A} \mathbb{E}\left[\sum_{t=1}^{T} \ell_t(i)\right] \\
&= \mathbb{E}\left[\sum_{t=1}^{T} \sum_{i \in A} p_t^v(i) \, \hat{\ell}_t^v(i)\right] - \min_{i \in A} \mathbb{E}\left[\sum_{t=1}^{T} \hat{\ell}_t^v(i)\right] \\
&\leq \frac{\ln K}{\eta(v)} + \frac{\eta(v)^2}{2} \mathbb{E}\left[\sum_{t=1}^{T} \sum_{i \in A} p_t^v(i) \, \hat{\ell}_t^v(i)^2\right]
\end{aligned}$$

as desired. □

## Proof of Lemma 2

**Lemma 2.** *Assuming agent $v$ uses the exponential-weights algorithm with a learning rate $\eta(v) \leq \frac{1}{2K}$, then for all $i \in A$:*

$$\left(1 - \eta(v) \, \hat{\ell}_t^v(i)\right) p_t^v(i) \leq p_{t+1}^v(i) \leq 2 p_t^v(i) .$$

*Proof.* From the exponential-weights update rule we have

$$\begin{aligned}
p_{t+1}^v(i) &= \frac{w_{t+1}^v(i)}{W_{t+1}^v} \\
&= \frac{W_t^v}{W_{t+1}^v} \exp\left(-\eta(v) \, \hat{\ell}_t^v(i)\right) p_t^v(i) \\
&\geq \exp\left(-\eta(v) \, \hat{\ell}_t^v(i)\right) p_t^v(i) && (W_{t+1}^v \leq W_t^v) \\
&\geq \left(1 - \eta(v) \, \hat{\ell}_t^v(i)\right) p_t^v(i) . && (1 - x \leq e^{-x})
\end{aligned}$$

as stated in the first inequality in the lemma. For the second inequality, note that

$$p_t^v(i) \, \hat{\ell}_t^v(i) = p_t^v(i) \frac{\ell_t(i)}{\mathbb{E}_t[B_t^v(i)]} B_t^v(i) \leq \frac{p_t^v(i)}{\mathbb{E}_t[B_t^v(i)]} \leq 1. \qquad (3)$$

Hence,

$$
\begin{aligned}
p_{t+1}^v (i) &= \frac{w_{t+1}^v (i)}{W_{t+1}^v} \\
&\leq \frac{w_t^v (i)}{W_{t+1}^v} \\
&= \frac{\sum_{j \in A} w_t^v (j)}{\sum_{j \in A} w_t^v (j) \exp \left( -\eta (v) \, \hat{\ell}_t^v (j) \right)} p_t^v (i) \\
&\leq \frac{\sum_{j \in A} w_t^v (j)}{\sum_{j \in A} w_t^v (j) \left( 1 - \eta (v) \, \hat{\ell}_t^v (j) \right)} p_t^v (i) && (1 - x \leq e^{-x}) \\
&= \frac{1}{1 - \eta (v) \sum_{j \in A} p_t^v (j) \, \hat{\ell}_t^v (j)} p_t^v (i) \\
&\leq \frac{1}{1 - \eta (v) K} p_t^v (i). && \text{(Eq. (3))}
\end{aligned}
$$

Assuming $\eta (v) \leq \frac{1}{2K}$, we obtain the desired bound. $\qquad\square$