[Reviews · NeurIPS 2019]

Reviewer 1



- It would be helpful to write down Exp3 more explicitly (at least using a numbered display) - it would be helpful to spell out the proposed cooperative MAB algorithm explicitly. It is somewhat hard to grasp how the current Alg. 1 and 4 relate to the proposed overall method. authors might wish to sharpen/reword the following: - "We can now obtain the same desired result.... " (what is the desired result?) - "The individual regret bound we introduced for the center-based..." (point to the bound e.g. using eqref) - ".and presented the center based cooperation policy." (point to the policy using eqref) - "This bound resolves an open question from [Cesa-Bianchi et al., 2019b] and also implies the result presented there" - "(i.e., agents are not partitioned to types.."

Reviewer 2



This paper is a follow up on the adversarial cooperative bandits problem. In this problem, N users pull at each time step the arm of their choice and receive an aversarial reward X_k(t) (common to all player pulling k). They can then communicate at each time step some information to their neighbors in a given communication graph. Cesa-Bianchi et al. provided a sublinear collective bound for this problem. This paper proposes an algorithm reaching a sublinear individual regret, answering to an open question. This paper presents a center based cooperative MAB algorithm. It first partitions the graph (in either centralized or decentralized setting) into connected components with leaders. The techniques used by this paper and the way they are combined are interesting and this paper directly answers to a question left open in a previous work. I feel that the first part of the paper is a direct extension of the paper of Cesa-Bianchi et al., which would not be consequent enough for a conference publication aside. I believe that the partitioning of the graph in the uninformed setting is a great complement to this first part. However, I find this part unclear and uncomplete right now. I trust the authors to improve this part for the rebuttal/camera ready version. --------------------------------------------------------------------- Major comments: 1. In my opinion the first part of the paper is a direct extension of the previous paper by Cesa-Bianchi et al. This is mainly because the proof of the regret relies on Lemmas 1 and 2 due to previous work, and as soon as these points hold, the proof is straightforward once we partition well the graph (which is done in the last part). For this reason, I believe that the construction of this graph is a significant content of the paper (especially in the uninformed setting). 2. We now come to my second point. The part on the uninformed setting is neither clear nor complete. I think the Luby's algorithm (in G^2) should be given in the Appendix. The line 2 of Algorithm 5 is totally unclear: how to run this algorithm on G^2|S_t ? We first need to be able to construct G^2|S_t. I also think that Lemma 11, Theorem 12 and Corollary 13 only hold with some high probabilty since Luby's algorithm is successful with probability 1-1/T. Lastly, the Centers-to-Components run in line 4 of Algorithm 5 might return nil Centers and U, and it does not give the distance from this center. I think this line as well should be written more carefully. 3. Also, I think it should be interesting to compare (on some classical graph structures) the difference in collective regret bound with the bound of Cesa-Bianchi et al. It would indeed be interesting to see how much we collectively lose to guarantee individual non regret strategies and how this behaves with the graph structure. ------------------------------------------------------------------------------- Minor comments 4. Page 1 Line 4, I think it would be better to keep the formula inline. 5. Page 1 Line 30: "We stress that ... the agent". This is also the kind of setting used in the multiplayer bandits problem, and this MAB problem could thus be mentioned in the related work. 6. At the first reading, I thought for a long time that what you called connected components were naturally given by the graph G. I only understood after a few pages that you created yourself the partition. Maybe you should rephrase it to insist on this point, e.g. by writing "We artificially partition ..." at Page 4 Line 129. 7. Page 4 line 145: you could also define d(c)=0 for completeness. 8. Page 6 algo 3 line 13: I think there is a typo on the subscript of U and M (t+1 instead of t, t instead of t-1) 9. Page 6 Line 192: There is a typo in the exponential (the d(v) should not appear) 10. Page 8, algo 5 line 2: why is there a K in the probability here ? This is linked with my major comment 2, as the probability is never mentioned in the following lemmas. Also, it is here unclear whether we look at (G|S_t)^2 or (G^2)|S_t. 11. About the regret bound, I believe it is possible to replace |N(v)| by 1+|N(v)|. In this case, we would retrieve the known bound for EXP3 in case of graph-independent agents. -------------------------------- The authors answer confirmed my belief that the clarity of the graph partitioning part will be improved for the camera ready version. Their answer thus comforted my score.

Reviewer 3



Originality: The paper deals with a well established problem and answers an open problem. It does so not just by analysing online algorithms but primarily through graph partitioning, which seems novel for the problem of cooperating agents. The algorithm (and notation!) for the bandit part of the paper is a direct continuation of the work of Cesa-Bianchi on cooperating agents, but it is the papers main focus to answer the open problem posed in that paper. Quality: Though proofs have not been checked, the paper seems sound. The approaches and analyses are well explained in the main text. Clarity: Though the technical material is well covered through lemmas in the main text, the paper is not well written as such. The are several places where it is not polished, and phrasings such as "... sharing of information makes perfect sense." (p. 1, l. 31) which are of little scientific value. In terms of improving on the cooperative setting, it is stated in the conclusion that the new bound implies the bound of Cesa-Bianchi. This should be explicit in the form of a corollary. Significance: As stated before, the problem is well motivated in literature and in general there is currently a significant interest in online learning on graphs. In general the paper has a very nice self contained scope, dealing with a single problem while still putting in significant amounts of new work. Especially the non technical part require a thorough review in the camera ready version. ==== After the author response: The authors have promised to address my concern in the camera ready version, but exactly how is not made explicit yet. My (accepting) score stands.

[Author Response · NeurIPS 2019]

First and foremost, we thank you for the positive reviews, and for taking your time to read and review our paper.

**Response to reviewer #1:**

- *"It would be helpful to write down Exp3 more explicitly"*:
  We will add a detailed explanation of Exp3 to the supplementary material.
- *"it would be helpful to spell out the proposed cooperative MAB algorithm explicitly"*:
  Our overall method is to run a partitioning algorithm (e.g. Algorithm 4) first, and use its result with the center-based policy (Algorithms 1 and 2). We will explain more clearly the interaction between the algorithms.
- *"authors might wish to sharpen/reword the following"*:
  We thank you for your rephrasing suggestions and we will implement them.
- *"Reorganize Sec. 3 and 4"*:
  We will reorganize the sections along the suggested lines.

**Response to reviewer #2:**

- *"convince me that the paper is very original in comparison with the paper of Cesa Bianchi et al."*:
  We completely agree the first part is a direct extension of [Cesa-Bianchi et al., 2019b], and is merely an introduction for what follows. Our main contribution is the graph partitioning methods, both in the informed and uninformed setting. Those methods are completely unrelated to the methods of [Cesa-Bianchi et al. 2019b], and they allowed us to solve the open individual regret problem from that paper.
- *"significantly improve the section 4.2"*, *"The part on the uninformed setting is neither clear nor complete"*:
  We will rewrite it more clearly and state everything explicitly. In addition, we will explain Luby's algorithm briefly in the main text, and also explicitly detail it in the supplementary material.
- *"We first need to be able to construct $\left(G^2\right)_{|S_t}$"*:
  We can run Luby's algorithm without explicitly constructing the graph, it just takes 2 steps to send a message instead of 1. As mentioned above, we will explain it in detail.
- *"Lemma 11, Theorem 12 ... only hold with some high probability"*:
  Lemma 11 and Theorem 12 indeed only hold with probability $1 - \frac{1}{T}$. We will mention that and add an explanation about it.
- *"Corollary 13 only hold with some high probability"*:
  This is not the case. Corollary 13 holds always (with probability 1), since we bound the **expected** regret. We will add a proof of this corollary to make it more clear.
- *"Algorithm 5 might return nil centers and $U$"*:
  We will explicitly state the case that Centers-to-Components returns nil.
- *"compare ... the difference in collective regret bound with the bound of Cesa-Bianchi et al"*:
  We stated in the paper (and we will add an explicit corollary to emphasize it) that we **strictly improve** Cesa-Bianchi et al. regret bound: our result implies theirs (asymptotically).
- *"Minor comments"*:
  We thank you for the suggestions and we will implement them. Specific comments:
  - *"Page 8, algo 5 line 2: why is there a $K$ in the probability here?"*:
    To answer your question as to why $1 - \frac{1}{TK}$ is the probability in Algorithm 5: it is because there are $K$ iterations, and we need the overall probability to be $1 - \frac{1}{T}$, so the expected regret would not increase.
  - *"it is here unclear whether we look at $\left(G_{|S_t}\right)^2$ or $\left(G^2\right)_{|S_t}$"*:
    We look at $\left(G^2\right)_{|S_t}$. We will change the notation to make it more clear.
  - *"replace $|\mathcal{N}(v)|$ by $1 + |\mathcal{N}(v)|$ ... retrieve the known bound for Exp3"*:
    If you notice, we actually defined $\mathcal{N}(v)$ to include $v$ itself, so we already retrieve the known bound for Exp3 like you suggested.

**Response to reviewer #3:**

- *"the non technical part require a thorough review in the camera ready version"*:
  We will polish and rephrase the less clear parts of the paper.
- *"it is stated ... the new bound implies the bound of Cesa-Bianchi. This should be explicit"*:
  We will add an explicit corollary and proof that we strictly improve the previous bound of Cesa-Bianchi et al..

[Meta-Review · NeurIPS 2019]

This paper studies the adversarial cooperative bandits problem, previously studied by Cesa-Bianchi et al. In this problem, N users pull at each time step the arm of their choice and receive an aversarial reward X_k(t) (common to all player pulling k). They can then communicate at each time step some information to their neighbors in a given communication graph. Cesa-Bianchi et al. provided a sublinear collective bound for this problem. This paper proposes an algorithm reaching a sublinear individual regret, answering to an open question. The reviewers liked the paper, and thought the model is interesting and that the contribution is solid.